# SingleInsert: Inserting New Concepts from a Single Image into Text-to-Image Models for Flexible Editing

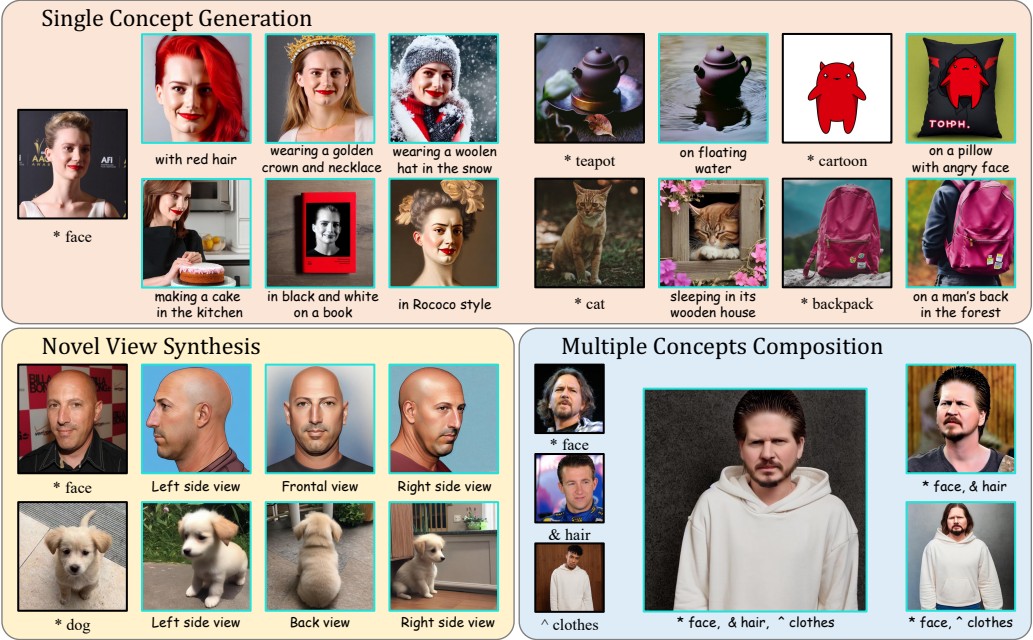

Figure 1: Three primary applications of our proposed **SingleInsert**: single concept generation, novel view synthesis, and multiple concepts composition (without joint training). The original images containing the intended concepts are in black boxes, while the editing results are in teal boxes.

## Abstract

Recent progress in text-to-image (T2I) models enables high-quality image generation with flexible textual control. To utilize the abundant visual priors in the off-the-shelf T2I models, a series of methods try to invert an image to proper embedding that aligns with the semantic space of the T2I model. However, these image-to-text (I2T) inversion methods typically need multiple source images containing the same concept or struggle with the imbalance between editing flexibility and visual fidelity. In this work, we point out that the critical problem lies in the foreground-background entanglement when learning an intended concept, and propose a simple and effective baseline for single-image I2T inversion, named SingleInsert. SingleInsert adopts a two-stage scheme. In the first stage, we regulate the learned embedding to concentrate on the foreground area without being associated with the irrelevant background. In the second stage, we finetune the T2I model for better visual resemblance and devise a semantic loss to prevent the *language drift* problem. With the proposed techniques, SingleInsert excels in single concept generation with high visual fidelity while allowing flexible editing. Additionally, SingleInsert can perform single-image novel view synthesis and multiple concepts composition without requiring joint training. To facilitate evaluation, we design an editing prompt list and introduce a metric named Editing Success Rate (ESR) for quantitative assessment of editing flexibility.

# 1 INTRODUCTION

Nowadays, some advanced text-to-image models (T2I models) (Rombach et al., 2022; Saharia et al., 2022; Shonenkov et al., 2023) have shown their sophistication in content creation. Given a random text prompt, these models are capable of generating diverse and high-quality results aligning well with the input semantics. Nonetheless, they usually fail to generate images containing a specific concept, which might be hard to describe in words or absent from the training set.

To tackle the above issue, existing methods (Gal et al., 2022; Ruiz et al., 2023a) try to optimize an embedding from the T2I model's semantic space to reconstruct the source images. These methods typically use 3-5 images encompassing varied conditions of the desired concept. However, if only a single image is given, the learned embedding tends to overfit the entire image rather than the intended concept, leading to severe editability degeneration.

Recently, various methods try to insert the concept from a single image into pre-trained T2I models. To alleviate the mentioned overfitting problem, some methods (Gal et al., 2023; Wei et al., 2023; Arar et al., 2023) impose restrictions to bind the learned embedding with known class embedding, some (Valevski et al., 2023; Chen et al., 2023d) craft datasets to create variations of the source images, some (Xiao et al., 2023) adopt an "early stopping" strategy or perform random augmentation to the input (Ma et al., 2023a) during inference to extend editability. However, these strategies can be hard to control and produce results with imbalanced editing flexibility and visual fidelity.

So, how to prevent overfitting when inserting a new concept from a single image into a pre-trained T2I model? We argue that the core problem lies in the training objective. First, the intended concept (also denoted as foreground) is what we want to learn rather than the irrelevant background. One straightforward idea is to optimize an embedding to match the foreground. Specifically, we train an image encoder to invert the source image to corresponding text embedding. During training, we only calculate the reconstruction loss within the foreground mask (obtained offline by GroundingDINO (Liu et al., 2023) and SAM (Kirillov et al., 2023)) to optimize the image encoder. However, we find it insufficient to prevent the predicted embedding from entangling with the whole image. Despite the varied reproduced backgrounds, the results are prone to misalign with the semantics of the text prompt (see Fig. 5), indicating that the learned embedding also significantly impacts the background. To eliminate this effect, we propose another loss restricting the influence of the foreground embedding on the denoising result of the background. Besides, if finetuning the T2I model for better visual fidelity, there exists another problem called *language drift* (Ruiz et al., 2023a). It means that the class embedding involved in training prompts (e.g., "face" in Fig. 2) also shifts towards the source image, causing the loss of class prior knowledge and a degradation in editability (see Fig. 5). To overcome this issue, we devise a semantic loss to maintain the impact of the class embedding on denoising results. Integrating the proposed techniques, our method is capable of producing results that strike a harmonious balance between editing flexibility and visual fidelity.

Our method only requires a single source image to achieve flexible editing of the intended concept, therefore called **SingleInsert**. Empirically, we adopt a two-stage scheme for the best performance. In the first stage (also denoted as the inversion stage), we only train the image encoder to invert the input image to proper embedding that characterizes the foreground. In the second stage (also denoted as the finetuning stage), we finetune the T2I model along with the image encoder, further restoring the identity of the target concept without sacrificing editability. As shown in Fig. 1, except for single concept generation, SingleInsert can perform single-image novel view synthesis. SingleInsert also enables multiple concepts composition without the need for joint training. After training separately, SingleInsert can combine different learned concepts (e.g., face, hair, clothes) together for further editing. Besides, we devise an editing prompt list and scoring criterion, and propose a metric named **Editing Success Rate (ESR)** to assess the editing flexibility of I2T inversion methods.

# 2 RELATED WORKS

**Diffusion-based Text-to-Image Generation**. Diffusion models (Ho et al., 2020; Nichol & Dhariwal, 2021) have shown their prowess in generating diverse, exquisite images. For better control, text-to-image diffusion models (Rombach et al., 2022; Saharia et al., 2022; Shonenkov et al., 2023) (referred to as T2I models in the following) have become mainstream nowadays. GLIDE (Nichol et al., 2021) introduces classifier-free guidance into the diffusion process by replacing the class

label with text. Imagen (Saharia et al., 2022) adopts a larger text encoder for better semantic alignments. unCLIP (Ramesh et al., 2022) trains a diffusion prior model to bridge the gap between CLIP (Radford et al., 2021) text and image latents. Stable Diffusion (SD) (Rombach et al., 2022) proposes Latent Diffusion Model (LDM), which compresses pixel-level generation to feature-level, largely improving efficiency without compromising generation quality. The recent SOTA T2I model IF (Shonenkov et al., 2023) shares a similar architecture with Imagen, yielding results up to 1,024 resolution. Our work is based on SD for its computational efficiency and open-source nature.

**Diffusion-based Image-to-Text Inversion**. Although the above T2I models can generate diverse and high-quality results aligning well with the given prompt, they struggle to deal with concepts that are hard to describe or unseen ones absent from the training datasets. Given 3-5 images containing the same concept, Textual Inversion (Gal et al., 2022) tries to optimize the embedding of a rare token to reconstruct the source images. For better fidelity, Dreambooth (Ruiz et al., 2023a) finetunes the UNet of SD apart from the target embedding, while Custom Diffusion (Kumari et al., 2023) and SVDiff (Han et al., 2023) only finetunes a small portion of the parameters. DisenBooth (Chen et al., 2023a) designs a two-branch pipeline to extract the common object from multiple inputs. Perfusion (Tewel et al., 2023) proposes a "Key-Locking" mechanism to mitigate attention overfitting. Chen et al. (2023c); Li et al. (2023); Shi et al. (2023) learn from massive image-text pairs to reduce the need for single-instance finetuning. ViCo (Hao et al., 2023) designs an image cross-attention module to integrate visual conditions into the denoising process for capturing object semantics.

Recently, several methods try to invert specific concepts from a single image for broader applications. ELITE (Wei et al., 2023) adopts the embedding from the deepest layer of the CLIP image encoder to reduce the low-level disturbances. E4T (Gal et al., 2023) and DATEncoder (Arar et al., 2023) add a penalty to the embedding for deviating from the class embedding. UMM-Diffusion (Ma et al., 2023b) and PhotoVerse (Chen et al., 2023b) introduce dual-branch condition mechanism to balance visual fidelity and semantic alignment. Similarly, FastComposer (Xiao et al., 2023) omits the learned embedding for the first few denoising steps to generate semantic-aligned layouts. ProFusion (Zhou et al., 2023) introduces PromptNet and proposes fusion sampling to remove the need for regularization. DreamIdentity (Chen et al., 2023d) crafts an editing dataset based on celebrity identity. Some methods use cropped images (Valevski et al., 2023) or foreground segmentations (Xiao et al., 2023; Ma et al., 2023a; Jia et al., 2023) as input to eliminate the influence of irrelevant backgrounds. The mentioned strategies can alleviate the overfitting problem to some extent. However, their results still exhibit an imbalance in terms of editing flexibility and visual fidelity.

In comparison, we present a suite of simple and effective constraints to alleviate the overfitting problem. The foreground loss in Sec. 3.2 shares a similar idea with BreakAScene (Avrahami et al., 2023). However, we verify in Sec. 4.3 that the results are still not satisfying when only adopting the foreground loss. Combining the other two regularizations, the predicted embedding learns to restore the identity of the foreground while avoiding perturbing the denoising procedure of the background, significantly improving the editing flexibility. Another advantage of SingleInsert is that it allows multiple concepts composition without requiring joint training, which is beyond BreakAScene.

## 3 METHOD

### 3.1 PRELIMINARY

In our experiments, we adopt the pre-trained Stable Diffusion (Rombach et al., 2022) v1.5 as the base T2I model (denoted as SD). The model consists of three components: a VAE (Kingma & Welling, 2013) (encoder $E(\cdot)$ and decoder $D(\cdot)$), a CLIP (Radford et al., 2021) text encoder $\tau_\theta(\cdot)$, and a diffusion UNet (Ronneberger et al., 2015) $\epsilon_\theta(\cdot)$. During training, first, the encoder $E$ maps the input image $I$ to the latent space: $x_0 = E(I)$. Second, a random timestep $t$ ($t \in [1, 1000)$) and noise $\epsilon$ ($\epsilon \sim \mathcal{N}(0, 1)$) is sampled. Then, $x_t$ is calculated as a weighted combination of $x_0$ and $\epsilon$:

$$x_t = \sqrt{\bar{\alpha}_t} \cdot x_0 + \sqrt{1 - \bar{\alpha}_t} \cdot \epsilon, \tag{1}$$

where $\bar{\alpha}_t$ is an artificial hyper-parameter set in the DDPM (Ho et al., 2020) noise scheduler. With $x_t$, the timestep $t$, and the text condition $y$ as the inputs, the training loss is formed as below:

$$L_{SD} = \mathbb{E}_{x_0, y, \epsilon \sim \mathcal{N}(0,1), t}[\|\epsilon - \epsilon_\theta(x_t, t, \tau_\theta(y))\|_2^2], \tag{2}$$

where both the UNet $\epsilon_\theta$ and the text encoder $\tau_\theta$ are jointly optimized. During inference, a random noise is sampled and iteratively denoised before being decoded into the generated image.

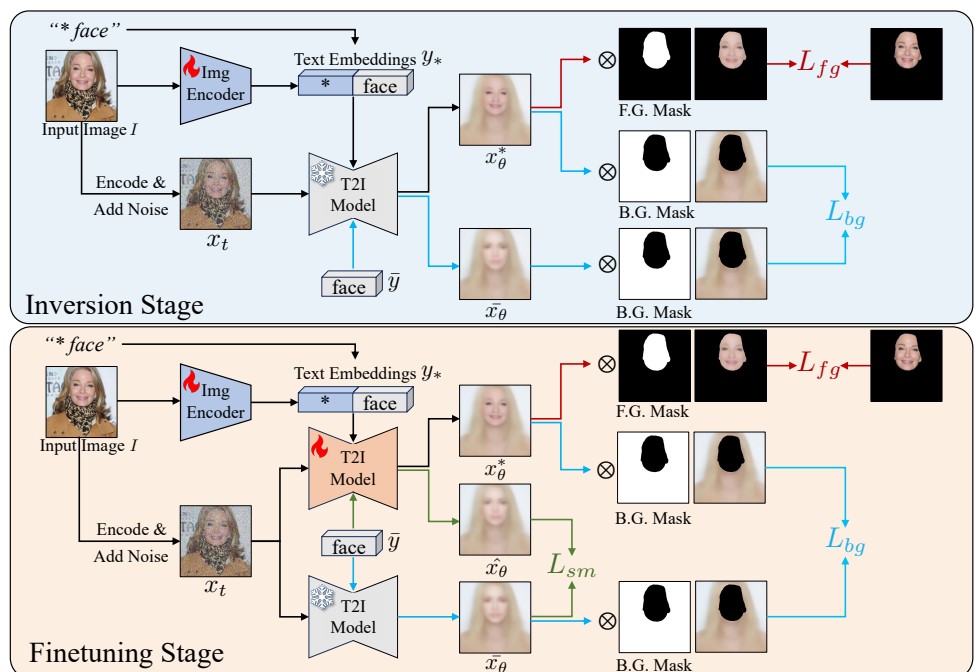

Figure 2: Illustration of our proposed SingleInsert framework. SingleInsert adopts a two-stage scheme. We fix the T2I model in the *Inversion Stage* (Stage I), and finetune the T2I model in the *Finetuning Stage* (Stage II). (Best viewed in color.)

## 3.2 INVERSION STAGE

In the inversion stage (Stage I), our goal is to find a proper embedding that best describes the intended concept without finetuning the T2I model. As shown in Fig. 2, given a target concept ("face" here), we aim to invert the source image $I$ to a textual embedding (represented as "*") so that "* face" can faithfully restore the identity of the target "face".

Our pipeline in this stage consists of two branches. In the condition branch, we send the input image $I$ to the image encoder to get a single-word embedding prediction to substitute the initial embedding of "*". We denote the replaced prompt as $y_*$. In the denoising branch, the input image $I$ is fed to the encoder of SD to get its feature map $x_0$. $x_t$ is formed as a weighted combination of $x_0$ and a random noise $\epsilon$. Under the guidance of $y_*$ and timestep $t$, the UNet receives $x_t$ as input to predict the added noise $\epsilon_\theta$. The feature maps are visualized as the corresponding images for clarity in Fig. 2.

**Foreground Loss**. In most cases, the target concept (face, hair, clothes, etc.) is only a part of the source image. To prevent the learned embedding from reproducing the irrelevant background, we adopt a strategy similar to Avrahami et al. (2023) to utilize the foreground mask when computing the reconstruction loss. Specifically, we use GroundingDINO (Liu et al., 2023) and SAM (Kirillov et al., 2023) to obtain the foreground mask $M_f$ of the intended concept ("face" in Fig. 2). During training, we only calculate the mean squared error in the foreground area between the predicted $x_\theta^*$ and the ground-truth $x_0$:

$$L_{fg} = \mathbb{E}_{x_0, y_*, t}[m_f \cdot \|x_\theta^* - x_0\|_2^2],\quad\quad(3)$$

where $x_\theta^* = \frac{x_t - \sqrt{1-\bar{\alpha}_t}\epsilon_\theta}{\sqrt{\bar{\alpha}_t}}$, according to Eq. 1. And $m_f$ is obtained by resizing $M_f$.

**Background Loss**. With the foreground loss, the generated images do have diverse backgrounds. However, the results usually misalign with the provided text prompts when editing with the learned embedding (see Fig. 5). The cause is that although only the reconstruction loss of the foreground area is utilized to optimize the embedding, it still impacts the position of the foreground or other information (e.g., poses) in the denoising results to reduce $L_{fg}$ (see Fig. 5). Our intuition is to minimize the impact of the learned embedding on the background area. To do this, we propose the background loss, denoted as $L_{bg}$. As in Fig. 2, we use the class prompt ("face" here, denoted as $\bar{y}$) to condition the denoising process to get $\bar{x}_\theta$. We restrict the background area of $x_\theta^*$ with $\bar{x}_\theta$ to make

sure "*" has a minor influence when denoising the background:

$$L_{bg} = \mathbb{E}_{x_0, y_*, \bar{y}, t}[(1 - m_f) \cdot \|x_\theta^* - \bar{x_\theta}\|_2^2]. \tag{4}$$

Combining the above two losses, our method can effectively eliminate the overfitting problem in the inversion stage (see Fig. 5). In this stage, the overall training loss is as follows:

$$L = L_{fg} + \gamma \cdot L_{bg}, \tag{5}$$

where $\gamma$ is an artificial hyper-parameter. In the inversion stage, we set $\gamma = 1.0$ as default.

### 3.3 FINETUNING STAGE

For better fidelity, a large portion of methods (Ruiz et al., 2023a; Chen et al., 2023d; Kumari et al., 2023; Ruiz et al., 2023b) choose to finetune the T2I model apart from optimizing the target embedding. Our proposed foreground loss and background loss also work for the finetuning stage. However, as Dreambooth (Ruiz et al., 2023a) points out, another problem called *language drift* occurs when finetuning the T2I model. That means the semantics of the class word ("face" in Fig. 2) have also been changed during training, leading to editability degeneration. To alleviate this problem, Dreambooth generates class-specific datasets (100+ samples) according to the category of the concept to regulate the class embedding. Apparently, creating such a class-specific dataset for every concept is costly. Besides, the samples in the crafted dataset are treated the same way as the source image during training, vastly increasing the computational burden.

**Semantic Loss**. So, what causes *language drift*? We argue that the leading problem is when optimizing the foreground loss or reconstruction loss in Dreambooth, the class embedding also leans towards the source image, leading to the loss of class priors and degenerated editability. To tackle the above issue, our solution is quite simple. As shown in Fig. 2, our goal is to maintain the semantics of the class word "face" while optimizing the embedding of "*". Instead of directly restricting the class word embedding from deviating, we impose a more thorough and reasonable constraint, which minimizes the difference between denoising results of the open T2I model and its fixed version conditioning on the class prompt. Specifically, given the noisy latent $x_t$ and the class prompt $\bar{y}$ as condition, we denote the outputs of the open T2I model and the fixed T2I model as $\hat{x_\theta}$ and $\bar{x_\theta}$. Then, the proposed semantic loss is formed as the mean squared error between $\hat{x_\theta}$ and $\bar{x_\theta}$:

$$L_{sm} = \mathbb{E}_{x_0, \bar{y}, t}[\|\hat{x_\theta} - \bar{x_\theta}\|_2^2]. \tag{6}$$

In the finetuning stage, the overall training objective is as follows:

$$L = L_{fg} + \gamma \cdot L_{bg} + \eta \cdot L_{sm}. \tag{7}$$

In our experiments, we set $\gamma = \eta = 1.0$ as default in this stage.

## 4 EXPERIMENTS

### 4.1 EXPERIMENTAL SETTINGS

**Implementation Details**. Our experiments adopt SD (Stable Diffusion (Rombach et al., 2022)) v1.5 as the base T2I model. We use a ViT-B/32 (Dosovitskiy et al., 2020) model as the image encoder to invert the source image. In the inversion stage, we freeze the SD v1.5 model and only train the image encoder for 50 iterations per concept with a learning rate of 1e-4 and a batch size of 16. In the finetuning stage, we add rank-4 lora (Hu et al., 2021) layers to both the text encoder and UNet of SD v1.5, and optimize the parameters of these lora layers along with the image encoder for another 100 iterations with the learning rate of 5e-5 and a batch size of 4. All the experiments are conducted on a single Nvidia V100 GPU. Please refer to the Appendix for more details.

**Datasets**. There are three primary sources of data. For human faces, we randomly select 100 images from the CelebA (Liu et al., 2018) training set and crop the largest region containing the faces before resizing to $512 \times 512$. For common objects, we utilize the dataset from Dreambooth (Ruiz et al., 2023a). Unlike Dreambooth, we only randomly choose one image for each subject as input. We also collect some images from the web. Before training, we utilize GroundingDINO (Liu et al., 2023) and SAM (Kirillov et al., 2023) to obtain the foreground mask of the intended concept according to

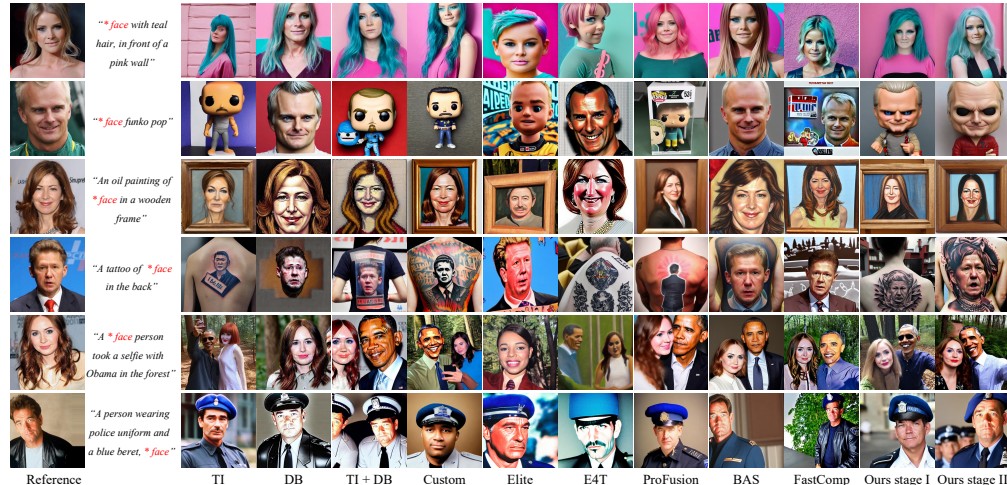

Figure 3: Qualitative comparisons on the "face" class.

its class prompt (face, hair, etc.). The prompt for each concept is set to "$* \langle class \rangle$", where "$*$" is the predicted embedding and "$\langle class \rangle$" stands for its corresponding class word.

**Evaluation Metrics**. To evaluate I2T methods, existing criteria mainly focus on three aspects: *image-alignment*, *text-alignment*, and *editing flexibility*. Following Dreambooth (Ruiz et al., 2023a), we adopt CLIP (Radford et al., 2021) visual similarity (CLIP-I) and DINO (Caron et al., 2021) similarity to evaluate the *image-alignment* degree, and CLIP text-image similarity (CLIP-T) as the measurement for *text-alignment*. However, some overfitting samples obtain higher *image-alignment* scores due to pixel-level consistency with the original image. As a solution, we introduce segmentation masks to achieve more accurate *image-alignment* evaluation. Specifically, we separately calculate the *image-alignment* scores on the foreground and background region. On the one hand, higher foreground score (CLIP-I-f, DINO-f) represents better visual fidelity of the intended concept. On the other hand, lower background score (CLIP-I-b, DINO-b) stands for a better disentanglement of the learned prompt from the undesired background information. As for *editing flexibility*, we adopt the average LPIPS (Zhang et al., 2018) to measure the diversity of samples under the same condition prompts (denoted as DIV). Moreover, to further evaluate the editability of each method, we devise ten complex text prompts, each containing one or two editing targets, and use these prompts as conditions to generate multiple samples according to different random seeds. Among these text prompts, we comprehensively consider common editing types, including background switching, abstract editing, attribute composition, foreground changing, multi-subject generation. Then, we calculate each method's editing success rate (denoted as ESR) as the measurement for the *editing flexibility*. We provide the details about metric ESR in the Appendix (Sec. A.2).

## 4.2 COMPARISONS

We compare our method with eight leading image-to-text inversion methods, including Textual Inversion (Gal et al., 2022), Dreambooth (Ruiz et al., 2023a), Custom Diffusion (Kumari et al., 2023), Elite (Wei et al., 2023), E4T (Gal et al., 2023), ProFusion (Zhou et al., 2023), BreakAScene (Avrahami et al., 2023) and FastComposer (Xiao et al., 2023). Among them, Elite, E4T, ProFusion, BreakAScene, and FastComposer are trained for single-image inversion. We evaluate all methods on the "face" class, while excluding E4T, ProFusion and FastComposer for concepts of other classes due to its imposed constraints. All the results are obtained by the official online demos (Elite, Fast-Composer) or official code (E4T, ProFusion, BreakAScene) or Diffusers library (Platen et al., 2022) (Textual Inversion, Dreambooth, Custom Diffusion) implementation.

**Qualitative Comparison**. As shown in Fig. 17, Textual Inversion (abbreviated as TI) (Gal et al., 2022) fails to preserve the identity of the target concept well. Dreambooth (abbreviated as DB) (Ruiz et al., 2023a) generates results with better fidelity at the cost of semantic alignments (row $2_{rd}$, $4_{th}$, $6_{th}$). The combination of TI and DB (TI+DB) produces better results compared to individual ones. However, some results also misalign with the semantics of the text prompts (row $4_{th}$, $5_{th}$, $6_{th}$). Similar to DB, Custom Diffusion (abbreviated as Custom) (Kumari et al., 2023) tends to learn from

Table 1: Quantitative comparison of different methods.

| Methods | CLIP-I-f ↑ | CLIP-I-b ↓ | DINO-f ↑ | DINO-b ↓ | CLIP-T ↑ | DIV ↑ | ESR ↑ |
|---|---|---|---|---|---|---|---|
| TI | 0.758 | 0.585 | 0.547 | 0.167 | 0.309 | 0.769 | 0.695 |
| DB | 0.748 | 0.578 | 0.532 | 0.160 | 0.270 | 0.754 | 0.515 |
| TI+DB | 0.769 | 0.613 | 0.519 | 0.180 | 0.296 | 0.751 | 0.480 |
| Custom | 0.679 | 0.571 | 0.465 | 0.173 | 0.297 | 0.742 | 0.510 |
| ELITE | 0.788 | 0.658 | 0.481 | 0.166 | 0.290 | 0.699 | 0.575 |
| E4T | 0.759 | 0.634 | 0.479 | 0.200 | 0.272 | 0.684 | 0.325 |
| ProFusion | 0.790 | 0.593 | 0.587 | 0.188 | 0.291 | 0.752 | 0.545 |
| BreakAScene | 0.835 | 0.698 | 0.603 | 0.233 | 0.273 | 0.722 | 0.375 |
| FastComposer | 0.819 | 0.658 | 0.534 | 0.177 | 0.266 | 0.748 | 0.405 |
| Ours (Stage I) | 0.822 | **0.555** | 0.582 | **0.152** | **0.317** | **0.776** | **0.845** |
| Ours (Stage II) | **0.857** | 0.601 | **0.609** | 0.176 | 0.311 | 0.753 | 0.825 |

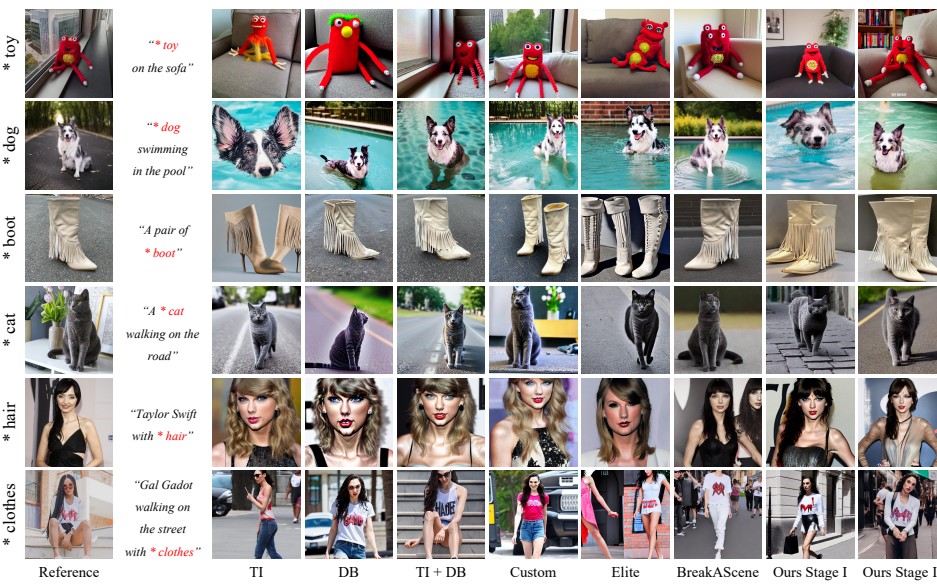

Figure 4: Qualitative comparisons in other categories.

the irrelevant background (the hair of row $3_{rd}$, the outfit of row $4_{th}$) and decrease the visual fidelity of the foreground (row $6_{th}$). E4T (Gal et al., 2023) shows over-saturated results (row $2_{rd}$, $3_{rd}$, $6_{th}$). ProFusion (Zhou et al., 2023) shows good editing flexibility at the cost of visual fidelity (row $2_{rd}$, $4_{th}$, $6_{th}$). Elite (Wei et al., 2023), BreakAScene (Avrahami et al., 2023), and FastComposer (Xiao et al., 2023) also utilize segmentation masks during training as our method does. Nonetheless, the results of these methods are more like the copy-and-paste version of the intended concept, which makes it hard for abstract editing (row $2_{rd}$, $3_{rd}$, $4_{th}$). In comparison, our method (stage I) generates results aligning well with the condition prompts while maintaining the fidelity of the target concept. After finetuning the T2I model (stage II), the visual consistency further improves while the editing outputs remain satisfying. As illustrated in Fig. 18, our method also shows superiority in concepts of other categories. To be noted, our method can handle complex concepts like hair and wearing clothes, which is difficult for existing methods (row $5_{th}$, $6_{th}$ in Fig. 18). For more qualitative results, please refer to the Appendix (Sec. A.6).

**Quantitative Comparison**. To conduct quantitative comparisons, we randomly select 10 images containing human "face". For each instance, we randomly assign a text prompt from our designed prompt list, and generate 10 samples according to different seeds. In total, we get $10 \times 10 = 100$ samples per method. We calculate the metrics mentioned in Sec. 4.1 and record the average scores of each criterion in Tab. 1. See Appendix (Sec. A.7) for quantitative comparisons on more categories.

As illustrated in Tab. 1, owing to the background regularization we propose, the learned embedding of our stage I best disentangles with the irrelevant background (CLIP-I-b, DINO-b), which effectively increases the editability of the embedding. Consequently, the results of our stage I show the best semantic consistency (CLIP-T) and editing flexibility (DIV, ESR). Besides, the visual consis-

Table 2: Ablation studies quantitative comparison. Here "baseline" stands for optimizing with the regular diffusion reconstruction loss (calculated over the whole image/feature map).

| Stages | Methods | CLIP-I-f ↑ | CLIP-I-b ↓ | DINO-f ↑ | DINO-b ↓ | CLIP-T ↑ | DIV ↑ | ESR ↑ |
|---|---|---|---|---|---|---|---|---|
| Stage I | Baseline | 0.879 | 0.855 | 0.792 | 0.564 | 0.202 | 0.472 | 0.190 |
| | $+L_{fg}$ | 0.938 | 0.840 | 0.740 | 0.457 | 0.227 | 0.532 | 0.170 |
| | $+L_{fg}, L_{bg}$ | 0.892 | 0.683 | 0.575 | 0.254 | 0.291 | 0.673 | **0.900** |
| Stage II | Baseline | **0.990** | 0.981 | **0.988** | 0.908 | 0.197 | 0.010 | 0.005 |
| | $+L_{fg}$ | 0.976 | 0.768 | 0.968 | 0.400 | 0.221 | 0.462 | 0.420 |
| | $+L_{fg}, L_{bg}$ | 0.909 | 0.591 | 0.600 | 0.124 | 0.251 | 0.744 | 0.655 |
| | $+L_{fg}, L_{bg}, L_{sm}$ | 0.944 | **0.465** | 0.761 | **0.102** | **0.316** | **0.771** | 0.810 |

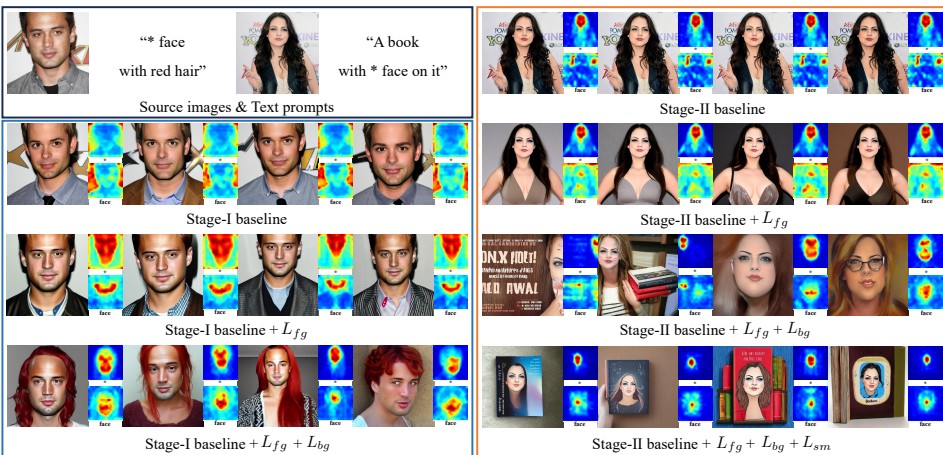

Figure 5: Ablation studies of the proposed three losses. We attach the cross attention heatmap of the embeddings of "*" and "face" (representing the learned concept) on the right side of each image. The red color stands for higher response, while blue indicates lower response.

tency (CLIP-I-f, DINO-f) of stage I also exceeds most existing methods. In stage II, the visual similarity of the target concept further improves, with slightly diminished editing flexibility, which corresponds to the qualitative results in Fig. 17 and Fig. 18. We also attach user study results in the Appendix, which reveals that our method aligns the best with human perception.

## 4.3 ABLATION STUDIES

We conduct extensive experiments to verify the effectiveness of our proposed losses, as shown in Fig. 5 and Tab. 2. Please refer to the Appendix (Sec. A.3) for more ablation studies.

**Foreground Loss**. From Fig.5 we can see that, when training with the foreground loss, the background area is more diverse and different from the original background. Moreover, the generated concept preserves more visual details in stage I, revealing that the optimization concentrates on the intended area. However, the editing results are not satisfying, misaligning the target prompts. The reason is that, despite removing the original background, the learned embedding still influences the background area. We can see that the learned embedding is correlated with the background in the man's heatmap and the body in the woman's heatmap.

**Background Loss**. In stage I, the editing results align with the target semantics well when adding the background loss. We can see from the heatmap that the learned embedding mainly correlates with the foreground and has little relevance with the background area, which shows the effectiveness of our proposed background loss in disentangling the background. This conclusion still holds in stage II. However, when finetuning with the above two losses, some challenging prompts' editing results are still unsatisfactory. As mentioned in Sec. 3.3, the cause of the editability degeneration lies in the *language drift* problem. As the heatmaps of the first three rows on the right side of Fig. 5 demonstrate, the corresponding area of the word "face" disperses when finetuning the T2I model.

**Semantic Loss**. To alleviate the *language drift* problem, we add the proposed semantic loss in stage II. As illustrated in Fig. 5, the editing results align well with the semantics when adding the semantic loss. The region of interest for the "face" embedding is more aggregated and concentrates

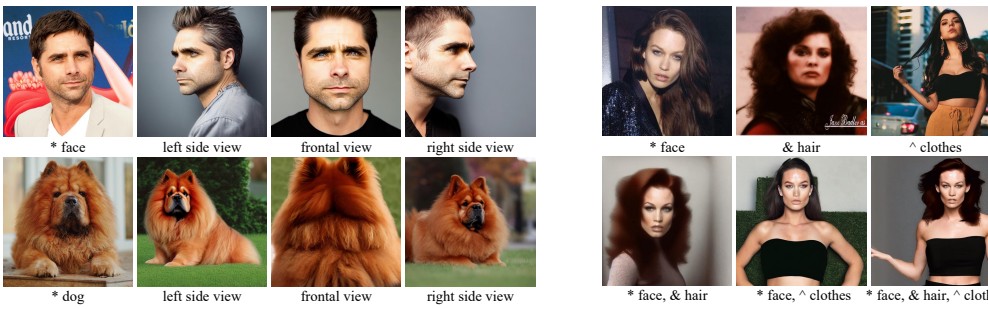

Figure 6: Novel view synthesis.          Figure 7: Multiple concepts composition.

on the foreground area. The quantitative evaluations in Tab. 2 also confirm the above conclusions. With the proposed three regularizations, our method achieves a good balance between visual fidelity and editing flexibility.

### 4.4 APPLICATIONS

Except for single concept editing, Our proposed SingleInsert enables other applications, such as novel view synthesis and multiple concepts composition. Please refer to the Appendix (Sec. refA.5) for more examples.

**Novel View Synthesis**. SingleInsert is capable of rendering the learned subject under novel viewpoints. As shown in Fig. 6, given a single image as input, our method can generate high-fidelity novel viewpoints according to text control, which is because SingleInsert utilizes copious class-specific visual priors in the T2I model to "dream up" reasonable novel views of the target concept.

**Multiple Concepts Composition**. Since our method effectively eliminates the influence of the learned concept on background areas, SingleInsert can combine multiple concepts of different classes together without the need for joint training. In Fig. 7, we show examples of two or three concepts composition when learning three concepts separately using SingleInsert. The proposed method handles different attribute compositions well, even for close-related ones like face and hair.

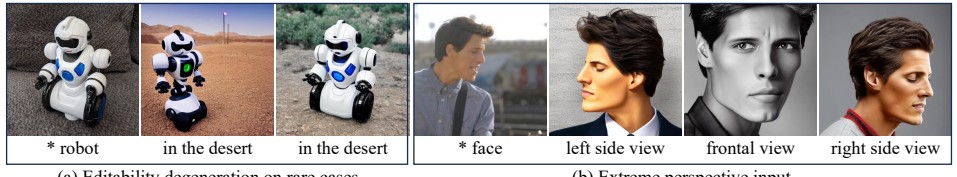

(a) Editability degeneration on rare cases          (b) Extreme perspective input

Figure 8: Two failure modes of SingleInsert.

### 4.5 LIMITATIONS

We demonstrate two main failure modes of our proposed SingleInsert in Fig. 8. First, when encountering rare cases of a class, SingleInsert sometimes generates results with an imbalanced trade-off between fidelity and diversity. The core problem is that the base T2I model has little prior knowledge of these rare cases or rare classes. Second, if the intended concept is captured from an extreme perspective, the synthesized novel viewpoints may be less visually similar to the intended concept.

## 5 CONCLUSIONS

In this paper, we propose SingleInsert, a method to insert new concepts from a single image into existing text-to-image models for flexible editing. The proposed regularizations of SingleInsert can effectively align the learned embedding with the intended concept while disentangling from the background. After obtaining the learned embedding, SingleInsert can generate flexible editing results with high visual fidelity. Besides, we showcase the capability of SingleInsert in synthesizing reasonable novel viewpoints from a single image and multiple concepts composition without joint training, revealing its vast potential for broader applications.

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

## A   APPENDIX

### A.1   DETAILS OF SINGLEINSERT

We demonstrate the proposed SingleInsert pipeline (Stage I and Stage II) in Alg. 1.

### A.2   DETAILS ABOUT EDITING SUCCESS RATE (ESR)

As mentioned in the main paper, there is currently a lack of quantitative metrics for measuring editing flexibility in the I2T inversion task. To this end, we propose to calculate the editing success rate (ESR) of each method given different prompts. For a more comprehensive measurement, we devise an editing prompt list that includes various common types of editing prompts as follows:

- · "A NBA/WNBA player, * face",
- · "* face funko pop",
- · "A tattoo of * face on the back",
- · "A person wearing police uniform and a blue beret, * face",
- · "A person wearing sunglasses smiling, * face",
- · "A side view * face with long/short hair",
- · "An oil painting of * face in a wooden frame",
- · "A * face person took a selfie with Obama in the forest",
- · "A book with a person wearing a crown on it, * face",
- · "* face with teal hair, in front of a pink wall",

---

**Algorithm 1** SingleInsert pipeline (Stage I and Stage II)

---

**Networks:**

$E$ — the $VAE$ encoder of $SD$

$\epsilon_\theta$ — the denoising $UNet$ of $SD$ (open)

$\epsilon_0$ — the denoising $UNet$ of $SD$ (fixed)

$\tau_\theta$ — the text encoder of $SD$

$E_i$ — the image encoder of $SingleInsert$

**Input:**

$I$ — the source image containing the intended concept

$M_f$ — the foreground mask of the intended concept

$\bar{y}$ — prompt : " $\langle$class$\rangle$ ", $\langle$class$\rangle$ represents the class word embedding

1: $x_0 = E(I), m_f = Resize(M_f),$ **//** Project the input image and mask to the feature space

2: $y_* = Concat(E_i(I), \bar{y}),$ **//** Prompt : " $* \langle$class$\rangle$ ", $*$ represents the predicted embedding

3: $t = Randint(1, 1000), \epsilon = Sample(N(0, 1)),$ **//** Randomly sample a timestep and noise

4: $x_t = \sqrt{\bar{\alpha}_t} \cdot x_0 + \sqrt{1 - \bar{\alpha}_t} \cdot \epsilon,$ **//** $\bar{\alpha}_t$ is a hyperparameter set in the noise scheduler

5: **Stage I:**

6: $\epsilon_\theta^* = \epsilon_0(x_t, t, \tau_\theta(y_*)), x_\theta^* = \frac{x_t - \sqrt{1 - \bar{\alpha}_t}\epsilon_\theta^*}{\sqrt{\bar{\alpha}_t}},$ **//** Compute $\mathrm{x}_\theta^*$

7: $\bar{\epsilon}_\theta = \epsilon_0(x_t, t, \tau_\theta(\bar{y})), \bar{x}_\theta = \frac{x_t - \sqrt{1 - \bar{\alpha}_t}\bar{\epsilon}_\theta}{\sqrt{\bar{\alpha}_t}},$ **//** Compute $\bar{\mathrm{x}}_\theta$

8: $L_{fg} = m_f \cdot \|x_\theta^* - x_0\|_2^2,$ **//** Calculate the foreground loss

9: $L_{bg} = (1 - m_f) \cdot \|x_\theta^* - \bar{x}_\theta\|_2^2,$ **//** Calculate the background loss

10: $L_{total} = L_{fg} + \gamma * L_{bf}.$ **//** Total loss

11: Update network $E_i$ to minimize $L_{total}$

12: **Stage II:**

13: $\epsilon_\theta^* = \epsilon_\theta(x_t, t, \tau_\theta(y_*)), x_\theta^* = \frac{x_t - \sqrt{1 - \bar{\alpha}_t}\epsilon_\theta^*}{\sqrt{\bar{\alpha}_t}},$ **//** Compute $\mathrm{x}_\theta^*$

14: $\hat{\epsilon}_\theta = \epsilon_\theta(x_t, t, \tau_\theta(\bar{y})), \hat{x}_\theta = \frac{x_t - \sqrt{1 - \bar{\alpha}_t}\hat{\epsilon}_\theta}{\sqrt{\bar{\alpha}_t}},$ **//** Compute $\hat{\mathrm{x}}_\theta$

15: $\bar{\epsilon}_\theta = \epsilon_0(x_t, t, \tau_\theta(\bar{y})), \bar{x}_\theta = \frac{x_t - \sqrt{1 - \bar{\alpha}_t}\bar{\epsilon}_\theta}{\sqrt{\bar{\alpha}_t}},$ **//** Compute $\bar{\mathrm{x}}_\theta$

16: $L_{fg} = m_f \cdot \|x_\theta^* - x_0\|_2^2,$ **//** Calculate the foreground loss

17: $L_{bg} = (1 - m_f) \cdot \|x_\theta^* - \bar{x}_\theta\|_2^2,$ **//** Calculate the background loss

18: $L_{sm} = \|\hat{x}_\theta - \bar{x}_\theta\|_2^2,$ **//** Calculate the semantic loss

19: $L_{total} = L_{fg} + \gamma * L_{bf} + \eta * L_{sm}.$ **//** Total loss

20: Update networks $E_i, \epsilon_\theta, \tau_\theta$ to minimize $L_{total}$

---

**Prompt Selection.** We choose the editing prompts according to these common types: background switching (in red text), abstract editing (in blue text), attribute composition (in green text), foreground changing (in orange text), and multi-subject generation (in brown text) (Best viewed in color).

**Scoring Criterion.** As shown in the proposed prompt list, each prompt has one or two editing goals. The editing is harder for those with only one editing goal since it requires global semantic consistency injection to the image. Each editing prompt equals one point, meaning those with two requirements are scored a half point for each target. Typically, we generate 10 images with different seeds per prompt and get $10 \times 10 = 100$ results for each method. Then, we calculate the average score as the final Editing Success Rate (ESR). The higher ESR suggests higher editing flexibility.

## A.3  MORE ABLATION STUDIES

### A.3.1  W/ OR W/O THE IMAGE ENCODER

In our experiments, we empirically find that utilizing an image encoder to map the source image leads to faster convergence than directly optimizing the embedding. We show the differences in training with or without an image encoder in Fig. 9 (both with the same batchsize and learning rate). As we can see, when trained with the same iterations, adding the image encoder produces better results with more visual detail restoration.

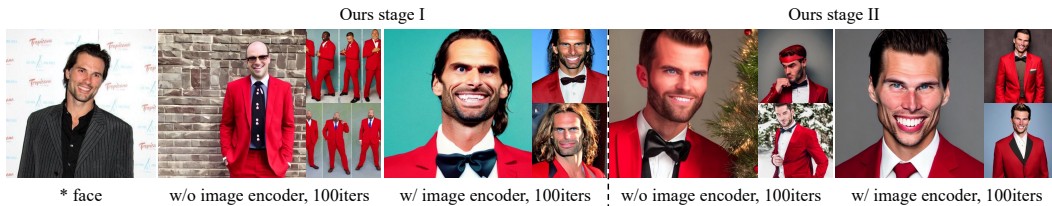

Figure 9: Training w/ or w/o the image encoder. The text prompt is "A * face man in red suits".

### A.3.2  W/ OR W/O THE CLASS PROMPT

During training, we use prompts like "* $\langle class \rangle$" ("$\langle class \rangle$" stands for the class prompt of the target concept, such as face, sunglasses, etc.) to represent the intended concept. To do so would have two advantages: First, the class prompt gives a good initialization, accelerating the embedding fitting procedure. Second, it utilizes the abundant class prior knowledge of the base T2I model, preventing the learned embedding from overfitting the foreground area. As illustrated in Fig. 10, the learned embedding loses some valuable characteristics, such as novel view rendering (the left part of Fig. 10) and concepts composition (the right part of Fig. 10), when training without the class prompt.

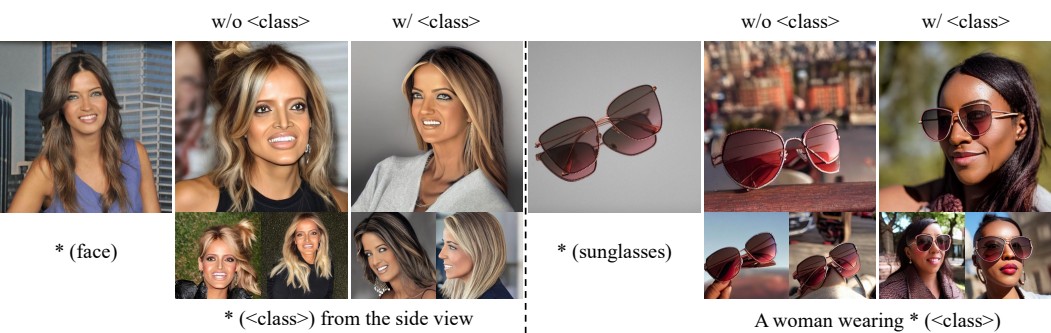

Figure 10: Training w/ or w/o the class prompt.

### A.3.3  W/ OR W/O STAGE I

As described in the main paper, we adopt a two-stage scheme for better performance. The inversion stage (Stage I) finds a proper embedding that produces visually similar samples while allowing

flexible editing. From Fig. 11, we can see that, when directly optimizing the T2I model along with the image encoder, it is hard to balance the fidelity preservation and the editability, even with more training iterations.

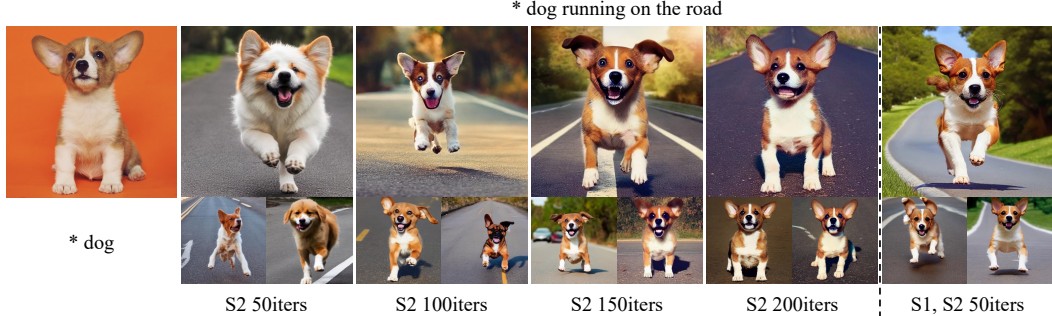

Figure 11: Training w/ or w/o Stage I. S1, S2 represent Stage I and Stage II, respectively.

### A.3.4 WEIGHT RATIO OF THE BACKGROUND LOSS AND FOREGROUND LOSS

The main paper demonstrates the individual effects of the proposed three losses. Among them, the foreground loss $L_{fg}$ concentrates the optimization on the foreground area, while the background loss $L_{bg}$ disentangles the learned embedding from the irrelevant background. In fact, the weight ratio of $L_{bg}$ and $L_{fg}$ plays a vital role in balancing the visual fidelity and editing flexibility. As shown in the upper half of Fig. 12, when the loss ratio of $L_{bg}$ and $L_{fg}$ is minor, the editing results are not satisfying, with fixed poses and misaligned hair colors. At the other extreme, when the loss ratio of $L_{bg}$ and $L_{fg}$ becomes too large, the editing results align well with the intended prompt but fail to preserve the identity of the target concept.

### A.3.5 WEIGHT RATIO OF THE SEMANTIC LOSS AND FOREGROUND LOSS

The semantic loss prevents the semantics of the known class prompt from changing during the finetuning stage. Moreover, the weight ratio of $L_{sm}$ and $L_{fg}$ controls the integration of the learned concept and editing background. As depicted in the lower half of Fig. 12, the foreground and background are isolated without $L_{sm}$. As the weight of $L_{sm}$ increases, the whole image is more natural and harmonious. However, too large the weight of $L_{sm}$ can also lead to a slight visual fidelity decrease.

### A.3.6 COMPARISON BETWEEN THE SEMANTIC LOSS AND EMBEDDING RECONSTRUCTION

We propose the semantic loss to alleviate the "language drift" problem. Instead of directly regulating the class embedding from changing, we pose a more thorough and reasonable constraint to preserve the semantics of the class prompt. We show the differences between these two strategies in Fig. 13. As we can see, directly reconstructing the class embedding shows worse editing results. We summarize the reason as follows: when reconstructing the embedding, the embedding similar to the class embedding will also *shift* for better restoration of the foreground area, resulting in semantic misalignment. In comparison, our proposed semantic loss directly regulates the denoising results and can better preserve the semantics of the known class embedding.

### A.3.7 CROPPED IMAGES AS INPUT

Our proposed SingleInsert mainly focuses on the overfitting problem of I2T inversion tasks. Instead of utilizing the foreground mask of the concept, some existing methods use cropped images as input to remove the irrelevant background as much as possible. We conduct experiments using the cropped foreground image as input and optimizing the image encoder to reconstruct the whole image. As shown in Fig. 14, even though the background area is minimal, the editing still fails in most cases, which reveals the superiority of our proposed techniques.

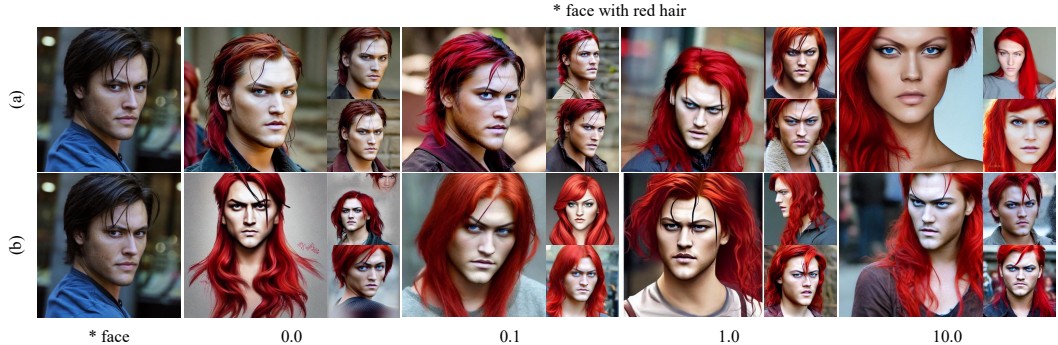

Figure 12: Ablation studies on the effect of loss weight ratio: (a) the loss ratio between $L_{bg}$ and $L_{fg}$, (b) the loss ratio between $L_{sm}$ and $L_{fg}$.

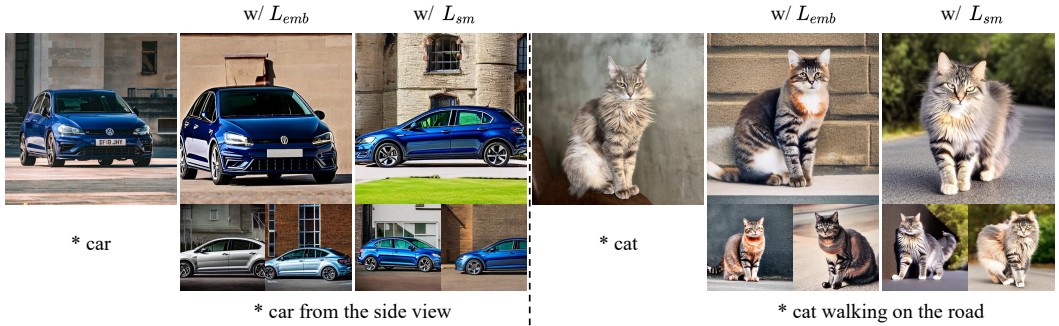

Figure 13: Comparisons of training with $L_{sm}$ or class embedding reconstruction loss (denoted as $L_{emb}$).

### A.3.8 DIFFERENT TYPES OF IMAGE ENCODER

In fact, the use of a ViT-B/32 image encoder is not necessary. Since the image encoder is inttroduced for faster convergence, we empirically found that we could replace the image encoder with a smaller CNN-based encoder with a slight performance degeneration. The comparison is presented in Tab. 3.

Table 3: Quantitative comparison on different types of image encoder.

| Image encoder | Params ↓ | Total Time ↓ | CLIP-I-f ↑ | CLIP-I-b ↓ | DINO-f ↑ | DINO-b ↓ | CLIP-T ↑ | DIV ↑ | ESR ↑ |
|---|---|---|---|---|---|---|---|---|---|
| ViT-B/32 | 88.4M | 3.5min | 0.857 | 0.555 | 0.582 | 0.152 | 0.317 | 0.776 | 0.845 |
| CNN | 11.7M | 5min | 0.839 | 0.560 | 0.568 | 0.153 | 0.320 | 0.783 | 0.820 |

### A.4 MORE APPLICATION EXAMPLES

Apart from single concept inversion, the proposed SingleInsert is capable of performing single-image novel view synthesis and multiple concepts composition (without joint training). We show more examples in Fig. 15 and Fig. 16. We compare the multiple concepts composition ability of our method against existing methods, the results are shown in Fig. 19.

To be noted, we only train different concepts in sequence before performing multiple concepts composition in the inference stage.

### A.5 MORE QUALITATIVE COMPARISONS

For comprehensive comparisons, we conduct experiments on more instances to compare the methods mentioned in the main paper. We show the qualitative comparisons in Fig. 17 (faces) and Fig. 18 (other categories). We provide three samples of every editing instruction to avoid randomness. We also compare our method with Subject Diffusion (Ma et al., 2023a), and show the results in Fig. 20.

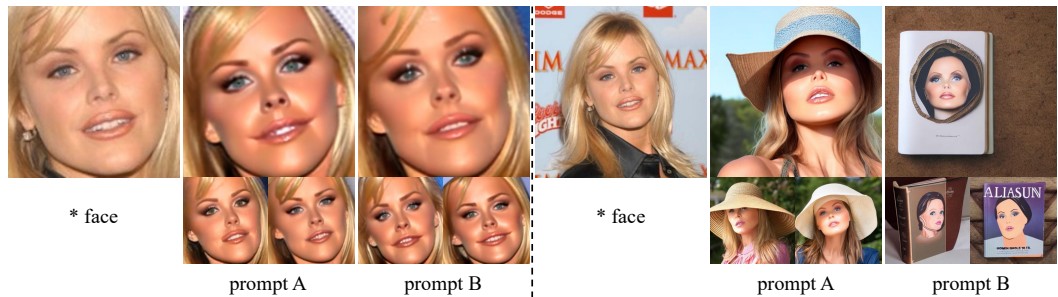

Figure 14: Cropped images as input *vs.* SingleInsert. Prompt A: "* face wearing a sun hat", prompt B:"* face on a book".

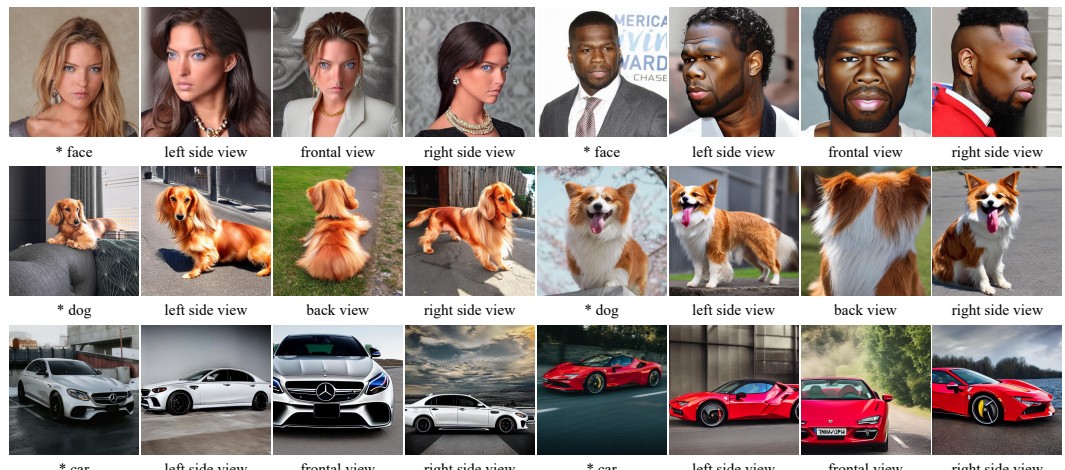

Figure 15: More novel view synthesis examples.

## A.6 MORE QUANTITATIVE COMPARISONS

In the main paper, we show the quantitative comparisons on the "face" class since one of the method is restricted to "faces". We also evaluate the performance of the compared methods on other categories. We calculate the average score of each metric on the samples shown in Fig. 18. The results are shown in Tab. 4. To be noted, although BreakAScene (Avrahami et al., 2023) achieves higher visual fidelity, it shows low editing flexibility. As shown in Fig. 18, the results of BreakAScene are more like the copy-and-paste version of the source image. We also add efficiency comparisons **??**, evaluations following Dreambooth (Ruiz et al., 2023a) 7, novel view synthesis comparison 5 for comprehensive evaluations.

Table 4: Quantitative comparison of different methods.

| Methods | CLIP-I-f ↑ | CLIP-I-b ↓ | DINO-f ↑ | DINO-b ↓ | CLIP-T ↑ | DIV ↑ |
|---|---|---|---|---|---|---|
| TI | 0.820 | 0.643 | 0.473 | 0.188 | 0.293 | 0.725 |
| DB | 0.829 | 0.705 | 0.541 | 0.247 | 0.282 | 0.663 |
| TI+DB | 0.874 | 0.752 | 0.570 | 0.275 | 0.279 | 0.652 |
| Custom | 0.824 | 0.662 | 0.540 | 0.222 | 0.293 | 0.715 |
| ELITE | 0.853 | 0.677 | 0.444 | 0.176 | 0.288 | 0.715 |
| BreakAScene | **0.920** | 0.745 | **0.686** | 0.295 | 0.276 | 0.525 |
| Ours (Stage I) | 0.842 | **0.639** | 0.496 | **0.167** | **0.297** | **0.744** |
| Ours (Stage II) | 0.898 | 0.655 | 0.629 | 0.230 | 0.296 | 0.681 |

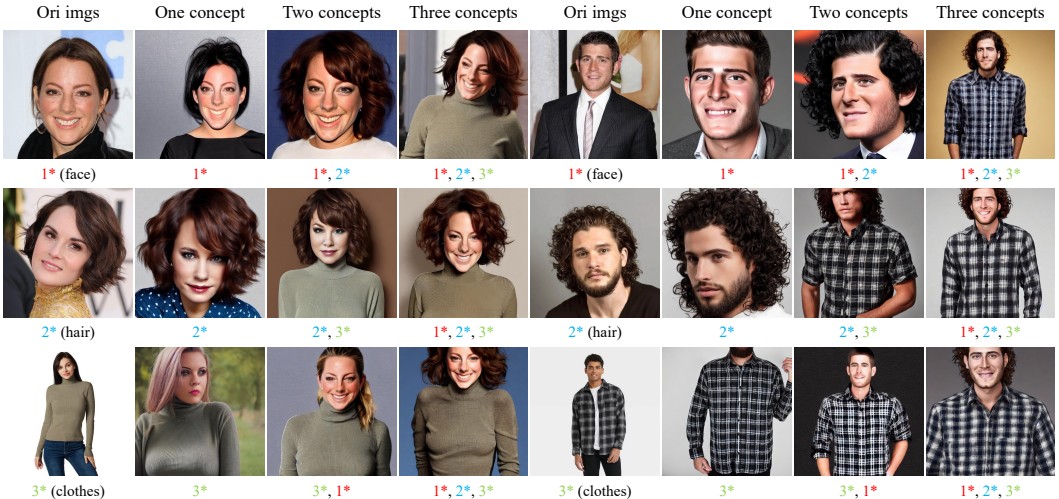

| Ori imgs | One concept | Two concepts | Three concepts | Ori imgs | One concept | Two concepts | Three concepts |

Figure 16: More multiple concepts composition examples.

Table 5: Quantitative comparison on novel view synthesis.

| View | Side | | | Back | | |
|---|---|---|---|---|---|---|
| Class | Human face | Car | Dog | Human face | Car | Dog |
| SD | **0.89** | **0.92** | **0.79** | / | **0.43** | **0.94** |
| TI | 0.72 | 0.69 | 0.69 | / | 0.38 | 0.84 |
| DB | 0.13 | 0.15 | 0.08 | / | 0.08 | 0.18 |
| Custom | 0.56 | 0.43 | 0.45 | / | 0.22 | 0.54 |
| Elite | 0.25 | 0.38 | 0.35 | / | 0.14 | 0.27 |
| FastComposer | 0.10 | 0.12 | 0.09 | / | 0.06 | 0.15 |
| BreakAScene | 0.37 | 0.45 | 0.39 | / | 0.15 | 0.43 |
| Ours | 0.79 | 0.83 | 0.72 | / | 0.40 | 0.80 |

## A.7 USER STUDY

We also include human evaluation, which is more representative in image-generation tasks. To do so, we invite 25 participants to choose their favorite editing results in consideration of three aspects: semantic alignment with the given prompt, visual similarity of the intended concept, and the quality of the generated image. We choose 10 "face" instances and 10 "other" instances and generate editing results according to 10 editing prompts, respectively. Overall, we get $20 \times 25 = 500$ votes in total. Then, we calculate the percentage of votes and show the human preference percentages in Tab. 9. As a result, our proposed SingleInsert generates results that are most in line with human perception.

Table 6: Finetuning efficiency comparison.

| Methods | Finetuning iters ↓ | Finetuning time ↑ | CLIP-I-f ↑ | CLIP-I-b ↓ | DINO-f ↑ | DINO-b ↓ | CLIP-T ↑ | DIV ↑ | ESR ↑ |
|---|---|---|---|---|---|---|---|---|---|
| TI+DB | 800 | 8min | 0.769 | 0.613 | 0.519 | 0.180 | 0.296 | 0.751 | 0.480 |
| E4T | **240** | **1min** | 0.759 | 0.634 | 0.479 | 0.200 | 0.272 | 0.684 | 0.325 |
| BreakAScene | 400 | 3.5min | 0.835 | 0.698 | 0.603 | 0.233 | 0.273 | 0.722 | 0.375 |
| Ours | 400 | 2min | **0.857** | **0.601** | **0.609** | **0.176** | **0.311** | **0.753** | **0.825** |

Table 7: Quantitative comparisons of different methods (following Dreambooth (Ruiz et al., 2023a)).

| Method | DINO ↑ | CLIP-I ↑ | CLIP-T ↑ |
|---|---|---|---|
| TI | 0.543 | 0.667 | 0.297 |
| DB | 0.623 | 0.759 | 0.270 |
| Custom | 0.637 | 0.773 | 0.276 |
| Elite | 0.656 | 0.769 | 0.261 |
| BreakAScene | **0.673** | **0.811** | 0.277 |
| Ours | 0.669 | 0.803 | **0.334** |

Table 8: Training time comparison of finetuning-based methods.

| Method | Total time |
|---|---|
| TI | 6min |
| DB | 8min |
| Custom | 10min |
| BreakAScene | 5.5min |
| Ours | **3.5min** |

Table 9: User studies of different methods.

| Class | TI | DB | TI+DB | Custom | Elite | BreakAScene | FastComposer | Ours S1 | Ours S2 |
|---|---|---|---|---|---|---|---|---|---|
| Faces | 0.040 | 0.036 | 0.056 | 0.080 | 0.020 | 0.056 | 0.156 | 0.164 | **0.392** |
| Others | 0.020 | 0.036 | 0.044 | 0.040 | 0.052 | 0.036 | \ | 0.160 | **0.612** |

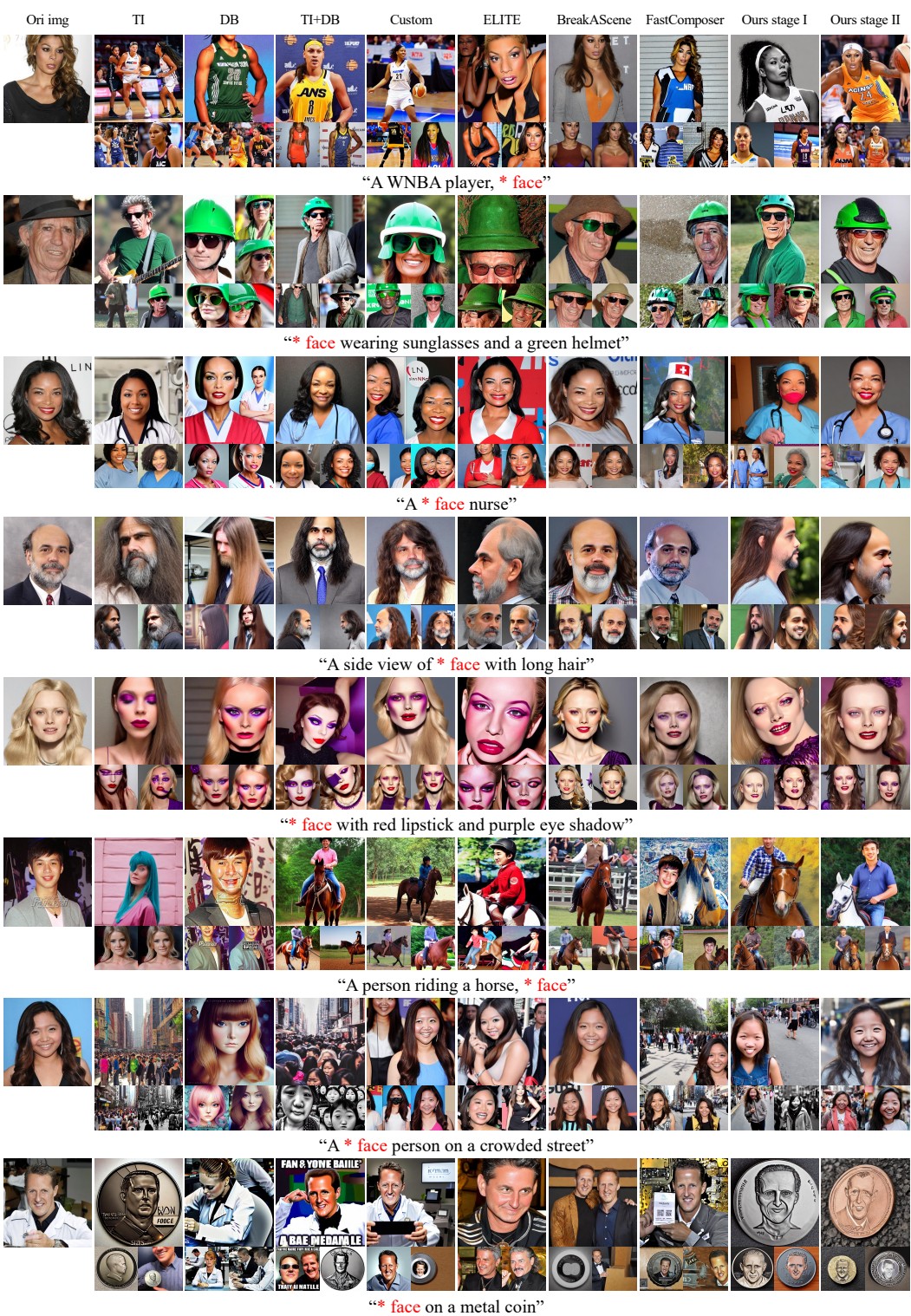

Figure 17: More qualitative comparisons on faces.

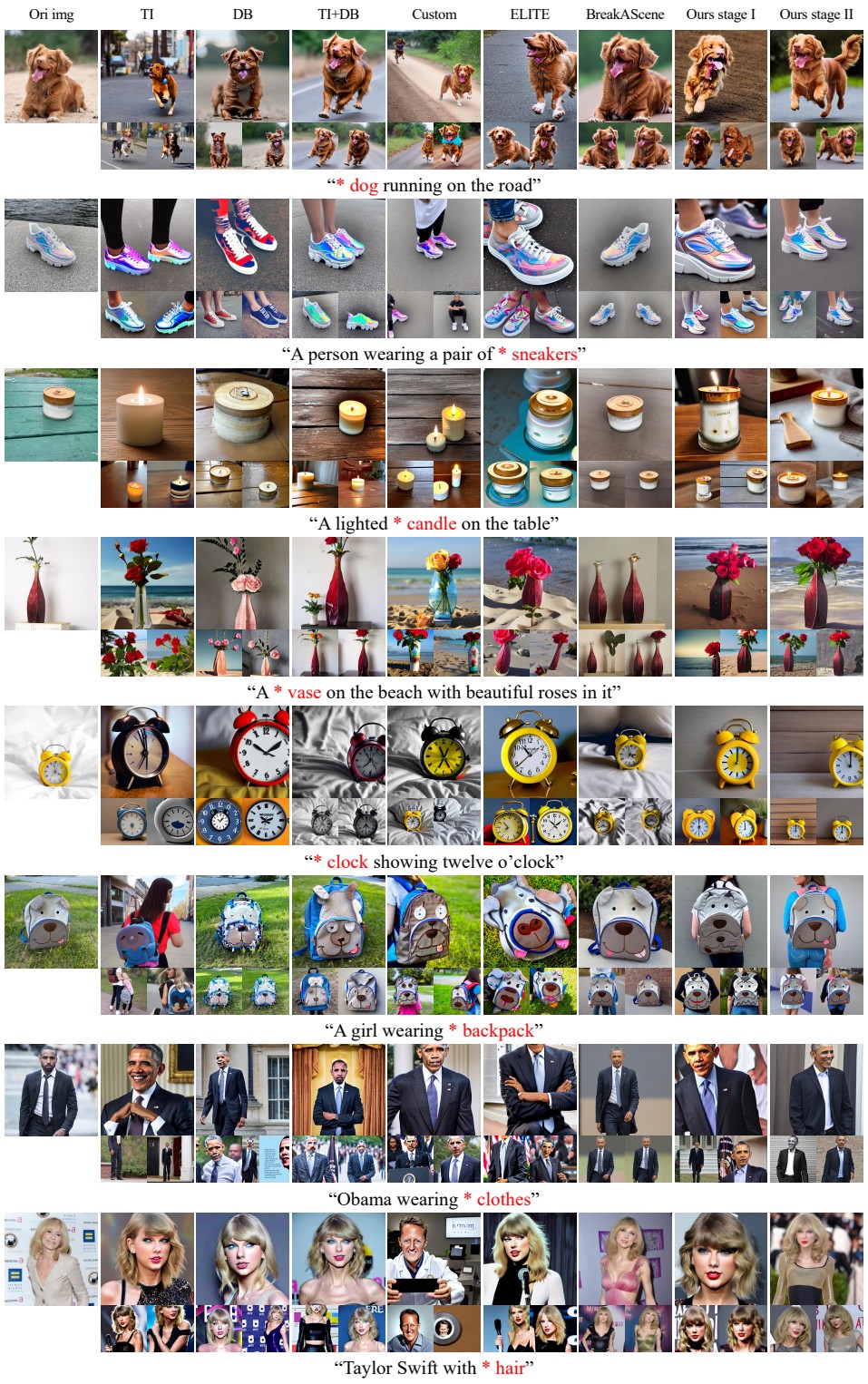

Figure 18: More qualitative comparisons on other categories.

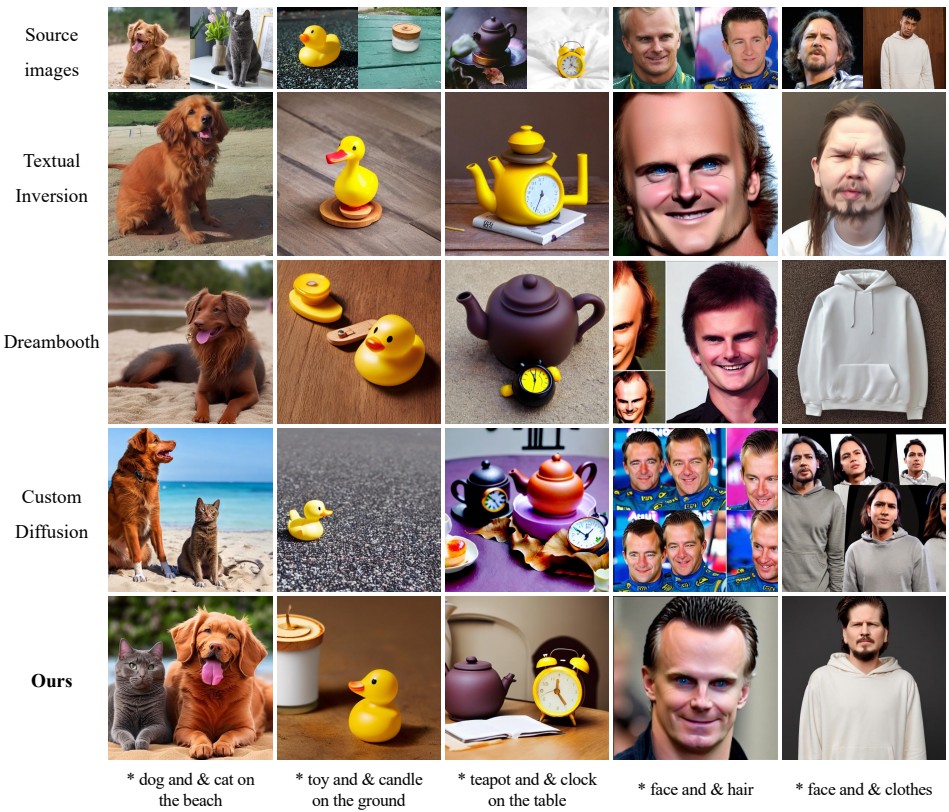

Figure 19: Multiple concepts inversion comparisons.

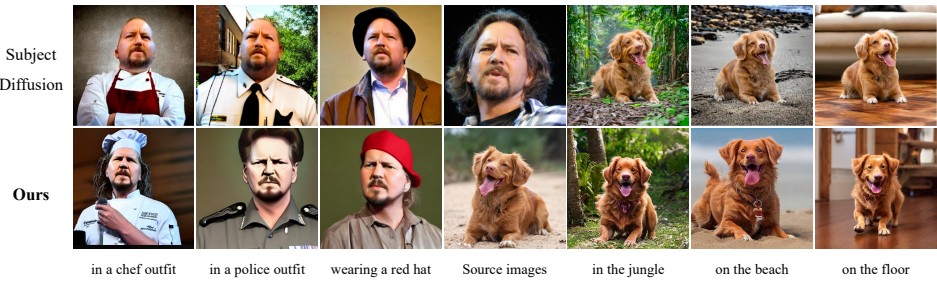

Figure 20: Comparison with subject diffusion.

