# OpenReview forum: "SingleInsert: Inserting New Concepts from a Single Image into Text-to-Image Models for Flexible Editing"
_ICLR.cc/2024/Conference — Submitted to ICLR 2024_

### Official Review · Reviewer_n7gt · 2023-10-30

**Soundness:** 2 fair
**Presentation:** 3 good
**Contribution:** 2 fair
**Rating:** 3
**Confidence:** 5

**Summary:**

This research addresses challenges in image-to-text (I2T) inversion and proposes "SingleInsert," a two-stage method that effectively separates foreground and background in learned embeddings. It enhances visual fidelity and flexibility in single-image concept generation, novel view synthesis, and multiple concept composition without joint training. The paper introduces the Editing Success Rate (ESR) metric for quantitative assessment of editing flexibility.

**Strengths:**

The paper is very easy to follow.

**Weaknesses:**

1. This paper seems to miss many related works or baselines.

- Taming encoder for zero fine-tuning image customization with text-to-image diffusion models.

- InstantBooth: Personalized Text-to-Image Generation without Test-Time Finetuning

- Enhancing Detail Preservation for Customized Text-to-Image Generation: A Regularization-Free Approach

2. Does hyper-dreambooth finetune t2i part?

3. The idea is not novel. Using mask to get more accurate object embedding is not new, and BG loss is largely used for this purpose also.

4. The two-stage training/finetuing plus the additional losses as restriction are more complicated than previous works, but lacking comparison with above related works.

5. According to the implementation details, the model requires retraining for each new concept, and in the finetuning stage, it relies on lora for better fidelity, which makes the soundness of the method even weaker.

6. The proposed ESR is worth more descriptions in the main part, since it is  a contribution.

**Questions:**

See above.

---

> ### Author Response · Authors · 2023-11-15
> **Response to Reviewer n7gt (part 1)**
>
> Thank you for the careful reading and helpful feedback. We respond below to your questions and concerns:
>
> **Q1:**
> Relevant papers.
>
> **R1:**
> Thanks for your reminder. We add the referred papers to related works in our revised paper. We find that all three mentioned papers[1,2,3] are released on Arxiv. In addition, we find that this paper [3] is also under the review process of ICLR 2024 like ours.
> As for comparisons, since [1,2] didn't release their code, we mainly add comparisons with [3]. Please refer to Sec. 4.2 in our revised paper, we add qualitative and quantitative comparisons with ProFusion[3] and also highlight the quantitative comparison with [3] in the table below. Both the qualitative and quantitative comparisons demonstrate the superiority of our method.
>
> | Method | CLIP-I-f ($\uparrow$)  | CLIP-I-b ($\downarrow$)  | DINO-f ($\uparrow$)  | DINO-b ($\downarrow$)  | CLIP-T ($\uparrow$) | DIV ($\uparrow$)| ESR ($\uparrow$) |
> | - | - | - | - | - | - | - | - |
> | ProFusion | 0.790 | 0.593 | 0.587 | 0.188 | 0.291 | 0.752 | 0.545 |
> | Ours (Stage I) | 0.822 | **0.555** | 0.582 | **0.152** | **0.317** | **0.776** | **0.845** |
> | Ours (Stage II) | **0.857** | 0.601 | **0.609** | 0.176 | 0.311 | 0.753 | 0.825 |
>
> [1] Jia X, Zhao Y, Chan K C K, et al. Taming encoder for zero fine-tuning image customization with text-to-image diffusion models[J]. arxiv preprint arxiv:2304.02642, 2023.
>
> [2] Shi J, Xiong W, Lin Z, et al. Instantbooth: Personalized text-to-image generation without test-time finetuning[J]. arxiv preprint arxiv:2304.03411, 2023.
>
> [3] Zhou Y, Zhang R, Sun T, et al. Enhancing Detail Preservation for Customized Text-to-Image Generation: A Regularization-Free Approach[J]. arxiv preprint arxiv:2305.13579, 2023.
>
> **Q2:**
> Question about HyperDreambooth.
>
> **R2:**
> HyperDreambooth directly predicts the LoRA weight of each instance and finetunes the LoRA weights in the finetuning stage.
>
> **Q3:**
> FG \& BG loss novelty.
>
> **R3:**
> First, As described in Sec. 3.2 (Foreground Loss) of our main paper, we do NOT claim that we are the first to introduce segmentation masks and apply the foreground loss in T2I inversion tasks. We have shown in ablations and Fig. 5 in our main paper that the results are not satisfying when only applying the existing foreground loss. It is worth noting that our other two proposed regularizations (BG loss and semantic loss) are proved to be essential for ensuring editing flexibility. Besides, at the end of Sec. 2, we discussed the relevance and differences of SingleInsert with BreakAScene[4], since it is the first paper to introduce the foreground loss into I2T inversion tasks. Second, as far as we know, there are no existing methods similar to our proposed background loss. We argue that ``BG loss is largely used for this purpose'' is not convincing.
>
> [4] Avrahami O, Aberman K, Fried O, et al. Break-A-Scene: Extracting Multiple Concepts from a Single Image[J]. arxiv preprint arxiv:2305.16311, 2023.
>
> **Q4:**
> About missing comparisons.
>
> **R4:**
> Please refer to the response of **R1**. Since [1,2] do not release their code, it is hard to have a comprehensive comparison with them. Moreover, Instantbooth[2] requires multiple source images as input, which differs from our single-image inversion setting.
> In addition, we follow your suggestion to add comparisons with ProFusion[3]. Please refer to Fig. 3 for qualitative comparisons and refer to Table 1 for quantitative comparisons in our revised paper.
> Both the qualitative and quantitative comparisons demonstrate the superiority of our SingleInsert method.

---

> ### Author Response · Authors · 2023-11-15
> **Response to Reviewer n7gt (part 2)**
>
> **Q5:**
> Method complexity.
>
> **R5:**
> To address your concern, we compare the finetuning
> efficiency of methods that have a finetuning stage in the table below (we multiply the batchsize and finetuning iterations as the whole iterations). All the inference is done on a single V100 GPU. From the table we can see that our method surpasses the other two methods in finetuning efficiency and achieves the best performance in quantitative evaluations, surpassing E4T by a large margin. We do not compare ProFusion here since it needs domain-specific pretraining.
>
> | Method | Finetuning iters ($\downarrow$) | Finetuning time ($\downarrow$) | CLIP-I-f ($\uparrow$) | CLIP-I-b ($\downarrow$) | DINO-f ($\uparrow$) | DINO-b ($\downarrow$) | CLIP-T ($\uparrow$) | DIV ($\uparrow$) | ESR ($\uparrow$)
> | - | - | - | - | - | - | - | - | - | - |
> | TI+DB | 800 | 8min | 0.769 | 0.613 | 0.519 | 0.180 | 0.296 | 0.751 | 0.480 |
> | E4T | **240** | **1min** | 0.759 | 0.634 | 0.479 | 0.200 | 0.272 | 0.684 | 0.325 |
> | BreakAScene | 400 | 3.5min | 0.835 | 0.698 | 0.603 | 0.233 | 0.273 | 0.722 | 0.375 |
> | Ours | 400 | 2min | **0.857** | **0.601** | **0.609** | **0.176** | **0.311** | **0.753** | **0.825** |
>
> In addition, we also compare the total training time with other single-source image finetuning methods in the table below. As shown in the table, our method surpasses other finetuning-based methods in training speed.
> We hope you will take these into consideration.
>
> | Method | Total time |
> | - | - |
> | TI | 6min |
> | DB | 8min |
> | Custom | 10min |
> | BreakAScene | 5.5min |
> | Outs | **3.5min** |
>
> **Q6:**
> About the two-stage finetuning pipeline and method soundness.
>
> **R6:**
> As for the two-stage finetuning pipeline, our method focuses on the task of single-image inversion, following the practice of some other finetuning-based methods (e.g., E4T, BreakAScene, ProFusion) in this research area, we also utilize a finetuning stage for better results. Please refer to the response of **R5**, it is worth noting that our SingleInsert demonstrates high finetuning efficiency among other finetuning-based methods.
>
> As for the method soundness, apart from the experiments in the original paper, we have tried our best to supplement the experiments in our revised paper, including qualitative and quantitative comparisons with E4T and ProFusion (Sec. 4.2 in main paper), comparisons with Subject-Diffusion using examples provided in their original paper (Fig. 20 in Appendix), comparisons of novel view synthesis and multiple-concepts inversion (Table 5 and Fig. 19 in Appendix) and more comprehensive comparisons of quantitative evaluations following Dreambooth's setting (Table 7 in Appendix). We hope these will address your concern about the soundness of our method.
>
> **Q7:**
> Question about the proposed metric ESR.
>
> **R7:**
> Thanks for your valuable suggestions!
> We add a brief introduction of the proposed ESR in our revised main paper (Sec. 4.1) and add more details about ESR in the Appendix (Sec. A.2).

---

### Official Review · Reviewer_i9TQ · 2023-11-01

**Soundness:** 3 good
**Presentation:** 3 good
**Contribution:** 2 fair
**Rating:** 5
**Confidence:** 4

**Summary:**

The paper proposes a new method for customized text-to-image generation, which considers disentanglement in learning the concept contained in user-provided image.

**Strengths:**

The proposed method tries to disentangle the influence of foreground and background in the given image, which is reasonable and straightforward.

Good results are presented in the paper, compared to related baselines.

Ablation studies are conducted, which help readers better understand the proposed method.

**Weaknesses:**

The proposed method seems to require more fine-tuning time compared to some related works (E4T only requires 5~15 steps, the proposed method requires 100 steps which is mentioned in section 4.1).

The idea of disentangling the foreground and background information has also been exploit in related works [1, 2]. Some of the related work have code publicly available online [2], but are not compared in this paper's experiments.

In quantitative evaluation, the authors didn't follow the setting in Dreambooth [3] to test the proposed methods on objects comprehensively. Specifically, the prompts used in the paper, on both human face and objects domain, may not be comprehensive enough. Dreambench proposed in [3] contains recontextualization, accessorization, and property modification prompts. On the contrary, example prompts shown in the paper are less comprehensive. Thus more comparisons are suggested.

Some related works also work on similar task with related ideas, which are suggested to be discussed in the paper.

[1]. DisenBooth: Identity-Preserving Disentangled Tuning for Subject-Driven Text-to-Image Generation. Hong Chen, Yipeng Zhang, Xin Wang, Xuguang Duan, Yuwei Zhou, Wenwu Zhu.


[2]. Subject-Diffusion:Open Domain Personalized Text-to-Image Generation without Test-time Fine-tuning. Jian Ma, Junhao Liang, Chen Chen, Haonan Lu.

[3]. DreamBooth: Fine Tuning Text-to-Image Diffusion Models for Subject-Driven Generation. Nataniel Ruiz, Yuanzhen Li, Varun Jampani, Yael Pritch, Michael Rubinstein, Kfir Aberman.

[4]. BLIP-Diffusion: Pre-trained Subject Representation for Controllable Text-to-Image Generation and Editing. Dongxu Li, Junnan Li, Steven C.H. Hoi.

[5]. PhotoVerse: Tuning-Free Image Customization with Text-to-Image Diffusion Models.  Li Chen, Mengyi Zhao, Yiheng Liu, Mingxu Ding, Yangyang Song, Shizun Wang, Xu Wang, Hao Yang, Jing Liu, Kang Du, Min Zheng.

[6]. Enhancing Detail Preservation for Customized Text-to-Image Generation: A Regularization-Free Approach. Yufan Zhou, Ruiyi Zhang, Tong Sun, Jinhui Xu.

[7]. InstantBooth: Personalized Text-to-Image Generation without Test-Time Finetuning. Jing Shi, Wei Xiong, Zhe Lin, Hyun Joon Jung.

**Questions:**

Can the authors provide more details about the data they collected from the web? Specifically, do those data consist of common object, human face, or both? What is the number of the collected samples?

In the fine-tuning stage, because a frozen T2I model is also used, how much extra memory do we need compared to the scenario without this model (both under LoRA setting).

Have the authors considered pre-training the model on a large-scale dataset? Will it reduce the fine-tuning time on testing images?

The author mentioned number of iterations needed, what is the actual total time needed in terms of seconds/minutes for customizing a new testing image?

---

> ### Author Response · Authors · 2023-11-15
> **Response to Reviewer i9TQ (part 1)**
>
> Thank you for the careful reading and helpful feedback. We respond below to your questions and concerns:
>
> **Q1:**
> Finetuning time.
>
> **R1:**
> Thanks for your concern. E4T[1] also adopts a two-stage scheme. However, there are two main reasons why we need more finetuning iterations than E4T. First, in our first stage, we do not finetune the base T2I model or addition LoRA layers, which intends to find an existing concept in the base model that corresponds to the target concept. In comparison, in the pretraining stage of E4T, it finetunes the weight offsets of T2I model using large-scale category-specific datasets, which extends the diversity of the original T2I model on specific class domain and gives a better class-specific initialization than the base T2I model could offer. However, the pretraining pipeline of E4T also limits their method to certain classes with abundant source images such as human faces, and costs much time for a single category. Second, as described in E4T, the batch size needs to be larger than 16 during finetuning stage, while our batch size is set to 4. In fact, we could also finetune for much fewer steps when setting the batch size to 16 without sacrificing the performance, which means the finetuning efficiency gap is not that large between our method and E4T. We compare the finetuning effeciency of methods that have a finetuning stage in the table below (we multiply the batchsize and finetuning iterations as the whole iterations). All the inference is done on a single Nvidia V100 GPU. From the table we can see that, our method surpasses the other two methods in finetuning efficiency, and achieves the best performance in quantitative evaluations, surpassing E4T by a large margin. We hope you may take these into consideration.
>
> |Method | Finetuning iters ($\downarrow$) | Finetuning time ($\downarrow$) | CLIP-I-f ($\uparrow$)| CLIP-I-b ($\downarrow$)| DINO-f ($\uparrow$)| DINO-b ($\downarrow$)| CLIP-T ($\uparrow$)| DIV ($\uparrow$)| ESR ($\uparrow$)|
> | - | - | - | - | - | - | - | - | - | - |
> | TI+DB | 800 | 8min | 0.769 | 0.613 | 0.519 | 0.180 | 0.296 | 0.751 | 0.480 |
> | E4T | **240** | **1min** | 0.759| 0.634| 0.479| 0.200| 0.272| 0.684| 0.325 |
> | BreakAScene | 400 | 3.5min | 0.835 | 0.698 | 0.603 | 0.233 | 0.273 | 0.722 | 0.375 |
> | Ours | 400 | 2min | **0.857** | **0.601** | **0.609** | **0.176** | **0.311** | **0.753** | **0.825** |
>
> [1] Gal R, Arar M, Atzmon Y, et al. Encoder-based domain tuning for fast personalization of text-to-image models[J]. ACM Transactions on Graphics (TOG), 2023, 42(4): 1-13.
>
> **Q2:**
> Relevant papers.
>
> **R2:**
> Thanks for your reminder! We add DisenBooth[2] in the related works. However, since Disenbooth requires multi-images as input, and it is not open-source yet, we can only discuss it in the revised paper. As for Subject-Diffusion[3], since the model is trained on a large-scale dataset privately owned by the authors, we are unable to reproduce the performance without the dataset or provided pretrained models. It is mentioned in its code repo's issue that their dataset or pretrained model can not be released due to relevant company restrictions. To address your concern, we compare our method with some examples shown in its paper. Please refer to Fig.~20 in the Appendix of our revised paper. We find that the learned concept in Subject-Diffusion appears to have fixed positions and poses, while our method achieves better editing flexibility. We add more qualitative comparisons against these methods in the revised paper.
>
> [2] Chen H, Zhang Y, Wang X, et al. DisenBooth: Disentangled Parameter-Efficient Tuning for Subject-Driven Text-to-Image Generation[J]. arxiv preprint arxiv:2305.03374, 2023.
>
> [3] Ma J, Liang J, Chen C, et al. Subject-diffusion: Open domain personalized text-to-image generation without test-time fine-tuning[J]. arxiv preprint arxiv:2307.11410, 2023.
>
> **Q3:**
> More related works.
>
> **R3:**
> Thanks for your valuable suggestion! We add the referred papers in our related works and compare the open source ones if accessible. Due to the popularity of the I2T inversion task, we found it hard to follow papers released on Arxiv very recently. But thank you so much! We definitely will add them to discussion.

---

> ### Author Response · Authors · 2023-11-15
> **Response to Reviewer i9TQ (part 2)**
>
> **Q4:**
> More quantitative evaluations.
>
> **R4:**
> Thanks for your reminder! We follow the setting of Dreambooth[4] and conduct extensive experiments on the dataset in Dreambooth to compare our method against others, the results are shown in the table below. To be noted, there is only a single source image containing the intended concept for a fair comparison. Although our visual similarity score (DINO and CLIP-I) is slightly lower than BreakAScene[5], our prompt fidelity score (CLIP-T) surpasses existing methods by a large margin, indicating the editing flexibility of SingleInsert, which is consistent with the conclusion in our main paper.
>
> | Method | DINO ($\uparrow$) | CLIP-I ($\uparrow$) | CLIP-T ($\uparrow$) |
> | - | - | - | - |
> | TI | 0.543 | 0.667 | *0.297* |
> | DB | 0.623 | 0.759 | 0.270 |
> | Custom | 0.637 | 0.773 | 0.276 |
> | Elite | 0.656 | 0.769 | 0.261 |
> | BreakAScene | **0.673** | **0.811** | 0.277 |
> | Outs | *0.669* | *0.803* | **0.334** |
>
> [4] Ruiz N, Li Y, Jampani V, et al. Dreambooth: Fine tuning text-to-image diffusion models for subject-driven generation[C]//Proceedings of the IEEE/CVF Conference on Computer Vision and Pattern Recognition. 2023: 22500-22510.
>
> [5] Avrahami O, Aberman K, Fried O, et al. Break-A-Scene: Extracting Multiple Concepts from a Single Image[J]. arxiv preprint arxiv:2305.16311, 2023.
>
> **Q5:**
> Online data details.
>
> **R5:**
> Thanks for your reminder! All the web images in our research are collected from [Pixel], shout out to their exquisite pictures. We collect 20 photos of models (10 male and 10 female) as sources for human faces and hairstyle and clothes. We also collect 10 car photos, 10 cat/dog photos as the source images of our novel view applications. The comparisons on common object categories are conducted using the mentioned data and Dreambooth[1] dataset.
>
> **Q6:**
> Extra memory consumption.
>
> **R6:**
> Thanks for your concern! During the finetuning stage, we add an extra T2I model to prevent language drift problems. Since the added T2I model is fixed during this stage, it consumes about 3.5GB vram since we loaded the model in FP16 precision. For the same purpose, many existing methods [4,5,6] double the batchsize to compute the prior preserving loss proposed in Dreambooth[4], which requires more memory.
>
> [6] Multi-Concept Customization of Text-to-Image Diffusion. Kumari, N., Zhang, B., Zhang, R., Shechtman, E., and Zhu, J. (2022).
>
> **Q7:**
> Large-scale dataset training.
>
> **R7:**
> Thanks for your question! We do have experimented on larger category-specific datasets such as human faces. We use 100, 1000, 10000 human face images to train our pretraining stage as described in the main paper as well. We empirically found that the performance of the first stage is close to the single-image setting. Since the max quality of the learned concept in the pretraining stage relies on the base T2I model, it could not reduce the finetuning time in test images, unless we also finetune the T2I model or LoRA layers in the pretraining stage. We choose single-image setting as default because our method does not restrict to certain classes as some relevant methods do. The lack of large-scale unusual class dataset is also one of our concerns.
>
> **Q8:**
> Total time.
>
> **R8:**
> Thanks for your reminder! We compare the training time with other finetuning methods in the table below. As shown in the table, our method surpasses other finetuning-based method in training speed. All the tests are conducted on a single Nvidia V100 GPU.
>
> | Method | Total time |
> | - | - |
> | TI | 6min |
> | DB | 8min |
> | Custom | 10min |
> | BreakAScene | 5.5min |
> | Outs | **3.5min** |

---

### Official Review · Reviewer_QGk3 · 2023-11-05

**Soundness:** 3 good
**Presentation:** 3 good
**Contribution:** 2 fair
**Rating:** 5
**Confidence:** 4

**Summary:**

This paper presents a two-stage Diffusion-based Image-to-Text Inversion algorithm that can mitigate overfitting when training with a single source image. It applies constraints to suppress the inversion of undesired background and the problem of language drift. Segmentation masks for foreground and background and predictions from the original diffusion model conditioned on the class of inversed concept are utilized to form the regularizations. It also designs an editing prompt list to quantitatively evaluate the edit flexibility of the inversed concept. With the proposed algorithm, in the non-trivial single-source-image scenario, this work achieves both high visual fidelity and editing flexibility, enabling novel view synthesis and multiple inversed concepts composition without joint training.

**Strengths:**

(1) The method allows more flexible ediitng for the inversed concepts from a single image, surpassing its baselines.
(2) The method presents a novel way to regularize the Image-to-Text Inversion process with predicted distributions by the original model.
(3) The paper presents an ediitng prompt list and a metric for quantitative evaluation of editing flexibility of inversed concepts.
(4) The paper clearly illustrates the motivations and the designs of the new proposed loss functions.
(5) The ablation studies clearly presents the value of each design of the proposed method.

**Weaknesses:**

(1) In section 4.4, the authors claim that the proposed approach enables single-image novel view synthesis. However, the experiments on this point are quite weak. Firstly, the algorithm cannot accurately control the viewpoint angle but can only control the view with text prompts "left side", "frontal", and "back side". Secondly, no evidence is provided to demonstrate how this constitutes an advancement compared to previous work on previous approaches. Thirdly, the generated novel view images also have drastic change on the background and even foreground appearance, which does not meet the requirement of novel view synthesis. Thus, I doubt that the claim of this contribution is not grounded.
(2) The application scenario of multiple concept composition is only demonstrated with a few examples but without comparison to previous work.
(3) On P6, section 4.1, a brief, if not detailed, introduction about the proposed metric ESR and the editing prompt list is expected to be given. The readers are supposed to have the basic idea about what is done in this evaluation after reading this section, instead of having to read the supplemental file to grasp it.

**Questions:**

(1) The proposed algorithm in this paper does not have a design specified for the single-source-image scenario and achieves single-source-image scenario by finetuning a large number of parameters, i.e. the whole T2I model and a ViT-B image encoder. So it would be natural to expect that the good performances generalize to the multiple-source-image scenario. Have you tried using the proposed algorithm for the multiple source-image inversion?
(2) Please refer to the questions in the weakness section.

---

> ### Author Response · Authors · 2023-11-15
> **Response to Reviewer QGk3**
>
> Thank you for the careful reading and helpful feedback. We respond below to your questions and concerns:
>
> **Q1:**
> Novel view applications claim.
>
> **R1:**
>
> We agree that the design of our method does not improve the accuracy of viewpoint control of the base T2I model. First, the ability of novel view synthesis of our method relies on the generation capacity of the T2I model. The main reason we claim the novel view synthesis application as one of our advantages is that: most of the previous single-image inversion methods suffer from foreground-background entanglement problems. The intended concept (e.g., human face) usually appears to show the same pose as in the source image (as in Fig. 17 in the revised paper). In contrast, our method enables flexible editing, with large variations, including poses, so that our proposed method can generate reasonable novel view of the intended concept with good visual fidelity. Second, to compare the advantages of our method in novel view synthesis against existing methods, we follow PrepNeg[1] to generate 100 samples each method according to viewpoint-specific prompts and compute the average success rate of viewpoint changing among the compared methods (All the concepts are captured from the frontal view. Due to the fact that human face does not have a backside, we choose to ignore it.). The results are attached below. From the table we can see that our method surpasses existing methods in viewpoint changing editing, though slightly worse than the base SD model, which indicates the superiority of our method. Third, as we focus on foreground concept, it is inevitable that the backgrounds can change with view point changed. We aim to improve this in the future and many thanks for you to point it out.
>
> | Method | Side face | Side Car | Side Dog | Back face | Back Car | Back Dog |
> | --- | --- | --- |  --- | --- | --- | --- |
> | SD  | **0.89** | **0.92** | **0.79** | / | **0.43** | **0.94** |
> | TI  | 0.72 | 0.69 | 0.69 | / | 0.38 | 0.84 |
> | DB | 0.13 | 0.15 | 0.08 | / | 0.08 | 0.18 |
> | Custom | 0.56 | 0.43 |  0.45 | / | 0.22 | 0.54 |
> | Elite  | 0.25 | 0.38 | 0.35 | / | 0.14 | 0.27 |
> | FastComposer  | 0.10 | 0.12 | 0.09 | / | 0.06 | 0.15|
> | BreakAScene | 0.37 | 0.45 | 0.39 | / | 0.15 | 0.43 |
> | Ours | *0.79* | *0.83* |  *0.72* | / | *0.40* | *0.80* |
>
>
>
> [1] Armandpour M, Zheng H, Sadeghian A, et al. Re-imagine the Negative Prompt Algorithm: Transform 2D Diffusion into 3D, alleviate Janus problem and Beyond[J]. arXiv preprint arXiv:2304.04968, 2023.
>
> **Q2:**
> Multiple concepts composition applications.
>
> **R2:**
> Thanks for your concern! We add more comparisons in the Appendix, please refer to Fig.~19 of our revised paper, which demonstrates that our SingleInsert achieves on par or even better results against existing methods in multiple concepts composition.
>
> **Q3:**
> Adding ESR description in main paper.
>
> **R3:**
> Thanks for your valuable suggestions!
> We add a brief introduction of the proposed metric ESR in our revised main paper (Sec. 4.1) and add more details about ESR in the Appendix (Sec. A.2). Regarding the editing prompt list, due to limited space in the main paper, we attach it in the Appendix (Sec. A.2). Please refer to our revised paper.
>
> **Q4:**
> Large finetune parameters
>
> **R4:**
> In fact, instead of finetuning the whole T2I model, we only finetune the LoRA layers, which contain about 0.75M parameters per concept. As for the image encoder, we have actually tried other image encoders such as a small CNN-based network (about 11.7M parameters), which produces results that are also appealing. We apply a ViT-B/32 image encoder simply for faster convergence. We follow the experiment setting in the main paper and report the quantitative evaluations in the table below. We have included these hyperparameters in the revised version.
>
> | Image encoder | Params ($\downarrow$) | Total time ($\downarrow$) | CLIP-I-f ($\uparrow$)| CLIP-I-b ($\downarrow$)| DINO-f ($\uparrow$)| DINO-b ($\downarrow$)| CLIP-T ($\uparrow$)| DIV ($\uparrow$)| ESR ($\uparrow$)|
> | - | - | - | - | - | - | - | - | - | - |
> | ViT-B/32 | **88.4M** | **3.5min** | **0.857** | **0.555** | **0.582** | **0.152** | 0.317 | 0.776 | **0.845** |
> | CNN | 11.7M | 5min | 0.839 | 0.560 | 0.568 | 0.153 | **0.320** | **0.783** | 0.820 |
>
> **Q5:**
> multiple source-image inversion.
>
> **R5:**
> Apart from the multiple-concepts composition examples shown in our original paper and supplementary materials, we add more examples in the revised paper (Fig. 16, Fig. 19 in Appendix), including comparison against other methods. We hope to bring it to your attention that our method can achieve comparable or even better results than other methods without the need for joint training in the multiple source-image inversion task.

---

> > ### Comment · Reviewer_QGk3 · 2023-12-05
> > **Thank authors for the rebuttal**
> >
> > Thank the authors for the response. I have read the author response which addressed some of my concerns. I would like to keep my original rating.

---

### Author Response · Authors · 2023-11-15
**To all reviewers and AC**

First of all, we thank all the reviewers for your insightful and constructive feedbacks. The valuable suggestions definitely make our work more solid and complete. We have tried our best to supplement the experiments and related works according to your concerns and advice. The revised paper mainly has the following differences:
1. We have added all the related works referred by the reviewers, and discuss them in the revised paper.
2. We have added qualitative and quantitative comparisons with E4T[1], ProFusion[2] in the main paper. Besides, we add comparisons with Subject-Diffusion[3] in the appendix using examples provided in their original paper since the pretrained model and private dataset of Subject-Diffusion are not open-source yet.
3. We have followed the suggestions by Reviewer QGk3 and Reviewer n7gt to add a brief but necessary introduction about the proposed ESR in the main part of the revised paper.
4. We have added comparisons of novel view synthesis and multiple-concepts inversion in the appendix of the revised paper.
5. We have added efficiency comparisons, image encoder ablation study, quantitative evaluations following Dreambooth[4] to provide more details of the proposed SingleInsert.
6. We have moved the content of supplimentary file to the appendix in the revised paper for easier reference.

Once again, we would like to express our gratitude to all the reviewers for your significant contributions in helping us improve our work. We have made every effort to address and implement the majority of the issues and suggestions raised. We hope to address any remaining concerns you may have.

[1] Gal R, Arar M, Atzmon Y, et al. Encoder-based domain tuning for fast personalization of text-to-image models[J]. ACM Transactions on Graphics (TOG), 2023, 42(4): 1-13.

[2] Zhou Y, Zhang R, Sun T, et al. Enhancing Detail Preservation for Customized Text-to-Image Generation: A Regularization-Free Approach[J]. ar**v preprint ar**v:2305.13579, 2023.

[3] Ma J, Liang J, Chen C, et al. Subject-diffusion: Open domain personalized text-to-image generation without test-time fine-tuning[J]. ar**v preprint ar**v:2307.11410, 2023.

[4] Ruiz N, Li Y, Jampani V, et al. Dreambooth: Fine tuning text-to-image diffusion models for subject-driven generation[C]//Proceedings of the IEEE/CVF Conference on Computer Vision and Pattern Recognition. 2023: 22500-22510.

---

### Author Response · Authors · 2023-11-21
**Expectations for feedback on the rebuttal process**

Dear Reviewers and ACs,

Firstly, I would like to express my gratitude for taking the time to carefully review our manuscript and provide valuable feedback. We deeply appreciate your insightful comments, and we have diligently addressed each of them in our rebuttal. Our aim was to provide clarifications and address any concerns raised during the review process.

However, it has been a few days since we submitted our rebuttal, and we have not received any further communication from you. Considering the time sensitivity of the paper review process, we are eagerly awaiting your response to our rebuttal so that we can further improve our research work based on your feedback.

If there is any additional information you require or if you have any questions regarding our rebuttal, we would be more than willing to provide further explanations and engage in a more detailed discussion concerning our paper.

We greatly appreciate your time and effort in reviewing our manuscript. We look forward to hearing back from you soon. Thanks.

---

### Meta-Review · Area_Chair_L72p · 2023-12-06

**Metareview:**

This paper proposes SingleInsert, which is a method for inserting new concepts from a single image into text-to-image generation models to enable flexible editing. Three reviewers reviewed the paper. The final ratings were 5, 5, 3. Positive points include the reasonable approach, some good results, and good ablation studies. Some common concerns include weak experimental evaluation or missing comparisons, incremental novelty, and some technical issues (e.g., finetuning time).  While the rebuttal addressed some of these concerns such as missing comparisons, it was not enough to convince the reviewers to increase their scores.  In the end, none of the reviewers were supportive of acceptance.  The paper, rebuttal, discussion, and author messages were carefully discussed among the ACs, and the ACs agree that the paper is not yet ready for publication and thus recommend rejection.  The ACs would like to encourage the authors to improve the paper and resubmit to another conference.

**Justification For Why Not Higher Score:**

The paper as it stands is not ready for acceptance, based on the points written above.

**Justification For Why Not Lower Score:**

N/A

---

### Decision · Program_Chairs · 2024-01-16

Reject